

# Ontogenetic, dietary, and environmental shifts in Mesosauridae

Antoine Verrière[1] and Jörg Fröbisch[1,2]

[1] Museum für Naturkunde, Leibniz-Institut für Evolutions- und Biodiversitätsforschung und Institut für Biologie, Berlin, Germany

[2] Institut für Biologie, Humboldt-Universität zu Berlin, Berlin, Germany

## ABSTRACT

Mesosaurs are the first secondarily aquatic amniotes and one of the most enigmatic clades of reptiles from the early Permian. They have long puzzled paleontologists with their unique morphologies: possessing an elongated skull with thin needle-like teeth, a long neck, large webbed hindlimbs, banana-shaped pachyosteosclerotic ribs, and a long tail. Here, we look at a large dataset of morphometric measurements from 270 mesosaur specimens in collections around the world. These measurements characterize skull, tooth, and limb proportions and their variation with size. This data presents evidence of surprising ontogenetic changes in these animals as well as new insights into their taxonomy. Our results support the recent hypothesis that *Mesosaurus tenuidens* is the only valid species within Mesosauridae and suggest that ''*Stereosternum tumidum*'' and ''*Brazilosaurus sanpauloensis*'' represent immature stages or incomplete specimens of *Mesosaurus* by showing that all three species occupy an incomplete portion of the overall size range of mesosaurs. Under the single-species hypothesis, we highlight a number of ontogenetic trends: (1) a reduction in skull length accompanied by an elongation of the snout within the skull, (2) an elongation of teeth, (3) a reduction in hind limb length, and (4) a reduction in manus length. Concurrent with these changes, we hypothesize that mesosaurs went through a progressive ecological shift during their growth, with juveniles being more common in shallow water deposits, whereas large adults are more frequent in pelagic sediments. These parallel changes suggest that mesosaurs underwent a diet and lifestyle transition during ontogeny, from an active predatory lifestyle as juveniles to a more filter-feeding diet as adults. We propose that this change in lifestyle and environments may have been driven by the pursuit of different food sources, but a better understanding of the Irati Sea fauna will be necessary to obtain a more definitive answer to the question of young mesosaur diet.

## INTRODUCTION

Mesosaurs are small marine parareptiles from the early Permian and one of the most enigmatic clades of early amniotes. Although they are exclusively found in the deposits of what was at the time a hypersaline inland sea extending over today's eastern South America and southern Africa (*Oelofsen & Araújo, 1983*; *Piñeiro et al., 2012b*; *Matos et al., 2017*; *Bastos et al., 2021*), and despite their short period of existence (*Soares, 2003*),

Corresponding author
Antoine Verrière,
antoine.verriere@mfn.berlin

mesosaurs represent a key snapshot in amniote evolutionary history. Indeed, they are the first secondarily aquatic amniotes, *i.e.,* the first fully terrestrial tetrapods to have return to live in water (*Carroll, 1988*). Yet their affinities are controversial, their origins are poorly known, and their lineage ended with them (*Laurin & Piñeiro, 2017*; *MacDougall et al., 2018*; *Ford & Benson, 2020*). Until recently, three monotypic genera were recognized: *Mesosaurus tenuidens* (*Gervais, 1869*), "*Stereosternum tumidum*" (*Cope, 1886*), and "*Brazilosaurus sanpauloensis*" (*Shikama & Ozaki, 1966*). Although these three taxa were originally supported by a number of diagnostic characters (*Shikama & Ozaki, 1966*; *Araújo, 1976*; *Oelofsen & Araújo, 1987*; *Modesto, 1999*; *Modesto, 2006*; *Modesto, 2010*), a recent review (*Piñeiro et al., 2021*) deemed these characters arbitrary and suggested *Mesosaurus tenuidens* to be the only valid species with "*Brazilosaurus*" and "*Stereosternum*" representing junior synonyms of *Mesosaurus*.

All mesosaurs present a striking morphology: a slender body with a very long tail, paddle-like hind limbs, a massive ribcage with thickened banana-shaped ribs, an elongated neck holding an elongate but very thin skull, and a long snout bearing thin needle-like teeth (Fig. 1). This peculiar appearance has resulted in conflicting hypotheses about their ecology, from amphibious (*Núñez Demarco et al., 2018*) to fully marine (*Modesto, 1999*; *Modesto, 2010*; *Canoville & Laurin, 2010*), from undulatory (*Braun & Reif, 1985*; *Villamil et al., 2016*) to partially limb-propelled swimmers (*Da Silva & Sedor, 2017*; *MacDougall et al., 2020*).

More than other aspects of their anatomy, the long thin teeth of mesosaurs have puzzled those who studied them the most. If they are somewhat reminiscent of those of other vertebrates such as dolphins or gharials, mesosaur teeth are singular in their length and their needle-like morphology. Consequently, they have no modern analog, which has led to contradicting attempts at characterizing the feeding habits of mesosaurs. Based on cranial morphology alone, *MacGregor (1908)* proposed a "fish"-based diet for mesosaurs. Subsequent studies discarded this interpretation (*Wiman, 1925*; *Araújo, 1976*), noting that their teeth were too fragile for active hunting and pointing to the absence of "fish" in mesosaur-bearing strata (*Oelofsen & Araújo, 1983*; *Piñeiro et al., 2012b*), although recent works show that fish were more frequent than previously thought (*Xavier et al., 2018*). In fact, based on their tooth morphology (*Pretto, Cabreira & Schultz, 2012*), the fauna of the Irati-Whitehill Sea (*Piñeiro et al., 2012b*; *Chahud & Petri, 2013*; *Xavier et al., 2018*), as well as gut content and coprolite analyses (*Silva et al., 2017*), most authors concluded that mesosaurs were filter-feeders trapping pygocephalomorph crustaceans in the net of their teeth (*Oelofsen & Araújo, 1983*; *Modesto, 1999*; *Modesto, 2006*; *Modesto, 2010*; *Piñeiro et al., 2012b*; *Silva et al., 2017*). Despite this, some authors suggested that mesosaurs could have been active hunters (*Pretto, Cabreira & Schultz, 2012*) or even scavengers (*Silva et al., 2017*).

Although interest and research effort into the paleobiology of mesosaurs has tremendously increased in the last decade and some aspects of mesosaur ecology have been addressed in the past, their lifestyle throughout ontogeny has not been fully investigated. Even though some authors have looked at the morphometrics of mesosaurs (*Araújo, 1976*; *Rossmann & Maisch, 1999*; *Rossmann, 2000*; *Rossmann, 2002*; *Núñez Demarco et al.,*

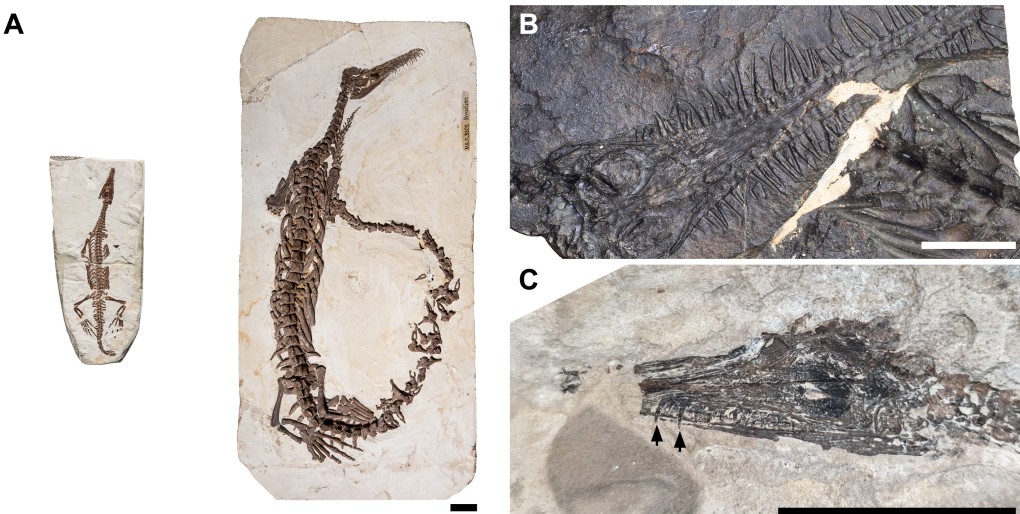

**Figure 1** **Comparison of morphologies in juvenile and adult mesosaurs.** Scale bars equal 20 mm. (A) Side-by-side comparison of a juvenile (BSPG 1979 I 37, on the left) and an adult "*Stereosternum tumidum*" (MB.R.5605, on the right) showing the ontogenetically changing proportions. (B) Close-up of the skull of adult *Mesosaurus tenuidens* NRM PZ R 207c showing the characteristic needle-like teeth of mesosaurs. (C) Close-up of juvenile "*Stereosternum tumidum*" GP/2E 9233 with arrows showing the shorter teeth of immature individuals.

*2018*; *Piñeiro et al., 2021*) or at histological growth marks (*De Ricqlès, 1974*; *Canoville & Laurin, 2010*; *Klein et al., 2019*), there have only been a few studies focusing on their ontogeny (*Piñeiro et al., 2012a*; *Piñeiro, Núñez Demarco & Meneghel, 2016*; *Núñez Demarco et al., 2018*; *Bickelmann & Tsuji, 2018*; *Verrière, Fröbisch & Fröbisch, 2021*; *Núñez Demarco, Ferigolo & Piñeiro, 2022*). This is surprising, as mesosaurs constitute a unique framework for understanding the complex links between diet and morphology in the fossil record. First, they comprise one of the few comprehensive ontogenetic series known in fossil amniotes (*Piñeiro et al., 2012a*; *Piñeiro, Núñez Demarco & Meneghel, 2016*; *Bickelmann & Tsuji, 2018*; *Núñez Demarco, Ferigolo & Piñeiro, 2022*). Second, they lived in a constrained environment with greatly reduced food variety, placing considerable limits on what their feeding habits could have been (*Soares, 2003*; *Piñeiro et al., 2012b*; *Xavier et al., 2018*). Lastly, they are the first secondarily aquatic amniotes, and understanding their dietary evolution has implications for the reconstruction of feeding habits in later marine amniotes.

Here, we provide the first quantitative assessment of ontogenetic changes in mesosaurs on such a large number of specimens. With this dataset, we test the hypothesis from *Núñez Demarco, Ferigolo & Piñeiro (2022)* that mesosaur grow isometrically. Our results support the single-species model of *Piñeiro et al. (2021)* by demonstrating that the formerly recognized species occupy incomplete segments of the mesosaur size range and show very little difference between them. Based on the single-species model, we do not retrieve the isometric growth measured by *Núñez Demarco, Ferigolo & Piñeiro (2022)* and instead show that the ontogeny of mesosaurs is marked by some significant allometric transformations and environmental partitioning, likely indicating a change in diet and lifestyle with growth.

## MATERIALS & METHODS

### Material investigated

We examined 270 articulated mesosaur specimens from collections around the world. Of these, 115 were previously attributed to *Mesosaurus tenuidens*, 96 to "*Stereosternum tumidum*", 13 to "*Brazilosaurus sanpauloensis*", and 46 were unidentified (Table S1). Of the identified specimens, 47 had a preserved skull and teeth, with 18 attributed to *Mesosaurus*, 29 to "*Stereosternum*", and only one to "*Brazilosaurus*". Earlier studies mentioned the presence of teeth in "*Brazilosaurus*" (*Rossmann, 2002*; *Modesto, 2006*; *Silva et al., 2017*), all of them relying on the average tooth length of 2 mm provided by *Rossmann (2002)*. We find this value contentious in different regards. First, *Rossmann (2002)* did not provide the raw measurements for his calculation. Second, we reexamined the material described by *Rossmann (2000)*; *Rossmann (2002)* as bearing teeth and noticed that elements he describes as teeth are in fact misidentified broken tooth sockets from the maxilla or the dentary (Fig. S1). Finally, out of the twelve specimens assigned to "*Brazilosaurus*" studied here, only ROM 28496 had some visible complete teeth. In all other specimens, teeth are either missing or completely broken, in any case not in a state of preservation allowing their measurement. Consequently, we must dismiss the average tooth length for "*Brazilosaurus*" of two mm given by *Rossmann (2002)*. For this reason, no specimens previously attributed to "*Brazilosaurus*" are included in our statistical analysis of teeth.

### Measurements

For each specimen, we measured the length of most preserved long bones: humerus, radius, ulna and metacarpals in the forelimb; femur, tibia, fibula, and metatarsals in the hind limbs. When possible, we also measured autopod length, from the proximal edge of the intermedium to the distal tip of digit III for the forelimb, and from the proximal edge of the astragalus to the distal tip of digit V in the hind limb, as digit III and V are the longest digits in the anterior and posterior autopod, respectively. In juveniles where carpals and tarsals were not ossified, the distalmost portions of the ulna or fibula were taken as most proximal extent of the autopodium. To account for cranial proportions, we measured the total skull length as well as antorbital and postorbital length.

Likely due to dental replacement, mesosaur teeth can greatly vary in size within a single specimen (*Modesto, 1999*; *Modesto, 2006*; *Piñeiro et al., 2021*). To better account for the dimensions of mesosaur teeth, we therefore chose not to use average tooth length and diameter, unlike previous works (*Piñeiro et al., 2021*). Instead, we measured the length of the longest visible tooth and the diameter of this tooth at its base.

Our specimens were preserved in various states of completeness, with some showing only the anterior or the posterior half of the body and others missing limbs or neck while others were mostly complete. Unlike *Núñez Demarco, Ferigolo & Piñeiro (2022)* who studied ontogenetic variation within bones, we wanted a shared proxy for specimen size to be able to model and compare variations in all bones independently. Thus, we calculated the average centrum length of dorsal vertebrae for each specimen, as this measurement has been shown to be a reliable estimate for body length (*Núñez Demarco et al., 2018*).

All measurements were taken using a caliper directly on specimens or Fiji 1.53c on distortion-corrected high-resolution photographs.

## Facies

Mesosaur specimens can be found in various deposits that represent different sections of the carbonate ramp boarding the Irati-Whitehill Sea (*Xavier et al., 2018*; *Ng, Vega & Maranhão, 2019*). To account for size distribution across deposit environments and potential niche partitioning of ontogenetic stages in mesosaurs, we categorized our specimens into five facies: (i) yellow limestone, (ii) white calcarenite, (iii) grey siltstone, (iv) grey mudstone, and (v) bituminous black shales. These facies represent a gradient in depth, energy level, and oxygenation of the water layers.

## Statistics

According to *Huxley (1932)*, simple allometry is described by the equation

$$y = kx^{\alpha}$$

or in its logarithmic form

$$\log y = \alpha \log x + \log k.$$

The allometric coefficient $\alpha$ characterizes the slope of the linear relationship between the logged valued of $y$ and $x$ and $\log k$ is the intercept of the linear relationship. Thus, an isometric growth (no changes in proportion to size during ontogeny) is characterized by an allometry coefficient of $\alpha = 1$ while values of $\alpha > 1$ and $\alpha < 1$ respectively reflect positive and negative allometry.

By transforming the simple allometry equation to express the ratio $y/x$ rather than $y$ as a function of $x$, we obtain

$$\frac{y}{x} = kx^{\alpha-1}$$

which in turns gives

$$\log \frac{y}{x} = (\alpha - 1)\log x + \log k.$$

In this equation we can define another allometry coefficient $\beta = \alpha - 1$. Thus, for the same values of $x$ and $y$, isometry is defined by $\beta = 0$ while negative and positive allometry are respectively characterized by negative and positive values of $\beta$, improving readability both graphically and numerically (*Klingenberg, 1998*). Therefore, we prefer $\beta$ to $\alpha$ in the present study.

For each metric, we computed a linear regression model of the logged ratio of the metric's values over size proxy value ($\log \frac{y}{x}$) as a function of logged size proxy values ($\log x$) and calculated the allometry coefficient $\beta$ to test for proportion changes throughout ontogeny. Similarity between species was also tested for each measurement. Linear modelling and statistical analyses were carried out in R 4.0.5 (*R. Core Team, 2021*).

## RESULTS

### Three-species hypothesis
#### *Skull*
Skull measurements behave differently with respect to size in the three formerly recognized mesosaur "species". In "*Brazilosaurus*", all three skull measurements show negative allometry, albeit not significant (Table 1). In *Mesosaurus*, postorbital length is the only skull metric to display a significant negative allometry, while total length and antorbital length both show a non-significant positive allometry (Table 1). In "*Stereosternum*", all three skull measurements show significant negative allometry but each with different allometry coefficients (Fig. 2). The negative allometry is less marked for antorbital length ($\beta = -0.090$, $p = 0.022$) than for total length ($\beta = -0.204$, $p < 0.001$), itself less marked than postorbital length ($\beta = -0.341$, $p < 0.001$).

For all three species, skull measurements rank similarly in terms of allometry coefficients with antorbital length having the highest, followed by total length and finally postorbital length last. This shows how skull elements vary differently in proportion throughout ontogeny. To better account for these variations, we measured the effect of allometry in function of total skull length instead of body size. In this configuration, we obtain very concordant results between the three species: antorbital length shows significant positive allometry, whereas postorbital length shows significant negative allometry (Table 2). Thus, over the course of ontogeny, the postorbital length decreases while the snout increases relative to overall skull length (Fig. 3).

#### *Teeth*
Several authors have noted an apparent change in tooth morphology between mesosaur species: "*Brazilosaurus*" and "*Stereosternum*" are described as bearing short, straight conical teeth whereas *Mesosaurus* is defined by thin, curved, and elongated teeth (*Oelofsen & Araújo, 1987*; *Rossmann, 2002*; *Modesto, 2006*). Indeed, for the same body size, adult *Mesosaurus* teeth are systematically longer than those of "*Stereosternum*" albeit similar in diameter (Figs. 2 and 3).

Tooth length and diameter in *Mesosaurus* both display positive allometry, although the linear relationship is not significant (Table 1). In "*Stereosternum*", both measurements show a significant negative allometry, with tooth diameter showing the lowest coefficient ($\beta = -0.622$, $p < 0.001$).

Since our results highlight variation in cranial dimensions throughout ontogeny, we also calculated allometry coefficients for tooth measurements with respect to skull length. Surprising, this shows non-significant negative allometry for tooth length in both species, whereas tooth diameter exhibits negative allometry for both species, although it is only significant in "*Stereosternum*" (Table 2). Thus, mesosaur teeth grow in length at the same rate as the skull, meaning that they do not become proportionally longer throughout ontogeny. However, these teeth keep their juvenile diameter and grow only in length in "*Stereosternum*" whereas both length and diameter grow isometrically to the skull in *Mesosaurus* (Fig. 4).

**Table 1    Allometry coefficient of logged size ratios for teeth and skull measurements.**

|  | Allometry coefficient $\beta$ | SE | $t$-value | $p$-value |
|---|---|---|---|---|
| **"B. sanpauloensis"** |  |  |  |  |
| Tooth length | – | – | – | – |
| Tooth diameter | – | – | – | – |
| Skull total length | −0.462 | 0.290 | −1.596 | 0.149 |
| Skull antorbital length | −0.509 | 0.362 | −1.405 | 0.198 |
| Skull postorbital length | −0.586 | 0.491 | −1.193 | 0.267 |
| *M. tenuidens* |  |  |  |  |
| Tooth length | 0.286 | 0.417 | 0.685 | 0.504 |
| Tooth diameter | 0.466 | 0.338 | 1.380 | 0.188 |
| Skull total length | 0.094 | 0.140 | 0.268 | 0.792 |
| Skull antorbital length | 0.204 | 0.160 | 1.279 | 0.222 |
| Skull postorbital length | **−0.511** | **0.275** | **−2.308** | **0.037** |
| **"S." tumidum** |  |  |  |  |
| Tooth length | **−0.381** | **0.115** | **−3.310** | **0.003** |
| Tooth diameter | **−0.787** | **0.096** | **−8.179** | **<0.001** |
| Skull total length | **−0.256** | **0.050** | **−5.131** | **<0.001** |
| Skull antorbital length | **−0.177** | **0.064** | **−2.770** | **0.008** |
| Skull postorbital length | **−0.376** | **0.074** | **−5.076** | **<0.001** |
| Single-species hypothesis |  |  |  |  |
| Tooth length | 0.029 | 0.149 | 0.191 | 0.849 |
| Tooth diameter | −0.653 | 0.106 | −6.190 | **<0.001** |
| Skull total length | −0.203 | 0.050 | −4.090 | **<0.001** |
| Skull antorbital length | −0.122 | 0.066 | −1.835 | 0.070 |
| Skull postorbital length | **−0.392** | **0.062** | **−6.296** | **<0.001** |

**Notes.**
Significant values ($p < 0.05$) are indicated in bold.

### *Limbs*

In all three species, most long bones show no significant difference in growth with isometry (Tables 3A–3C), meaning that their proportional length with respect to body size does not change throughout ontogeny. However, there are several exceptions to this.

In "*Brazilosaurus*", femur, tibia and metatarsal V length show significant negative allometry (Table 3A). Although this might reflect a size reduction of the posterior limb during ontogeny in this species, those measurements are only documented for eight adult specimens in our dataset and likely do not reflect the entire ontogenetic trajectory of the species.

In *Mesosaurus*, most forelimb bones exhibit a significant positive allometry, aside from metacarpals II, IV, and V (Fig. 2). While this suggests that the anterior autopod increases in size, there are very few juveniles or subadult *Mesosaurus* in our dataset. Thus, much like for "*Brazilosaurus*", *Mesosaurus* forelimb measurements are concentrated in the adult region of the morphospace, and do not satisfyingly account for the entire ontogenetic variation.

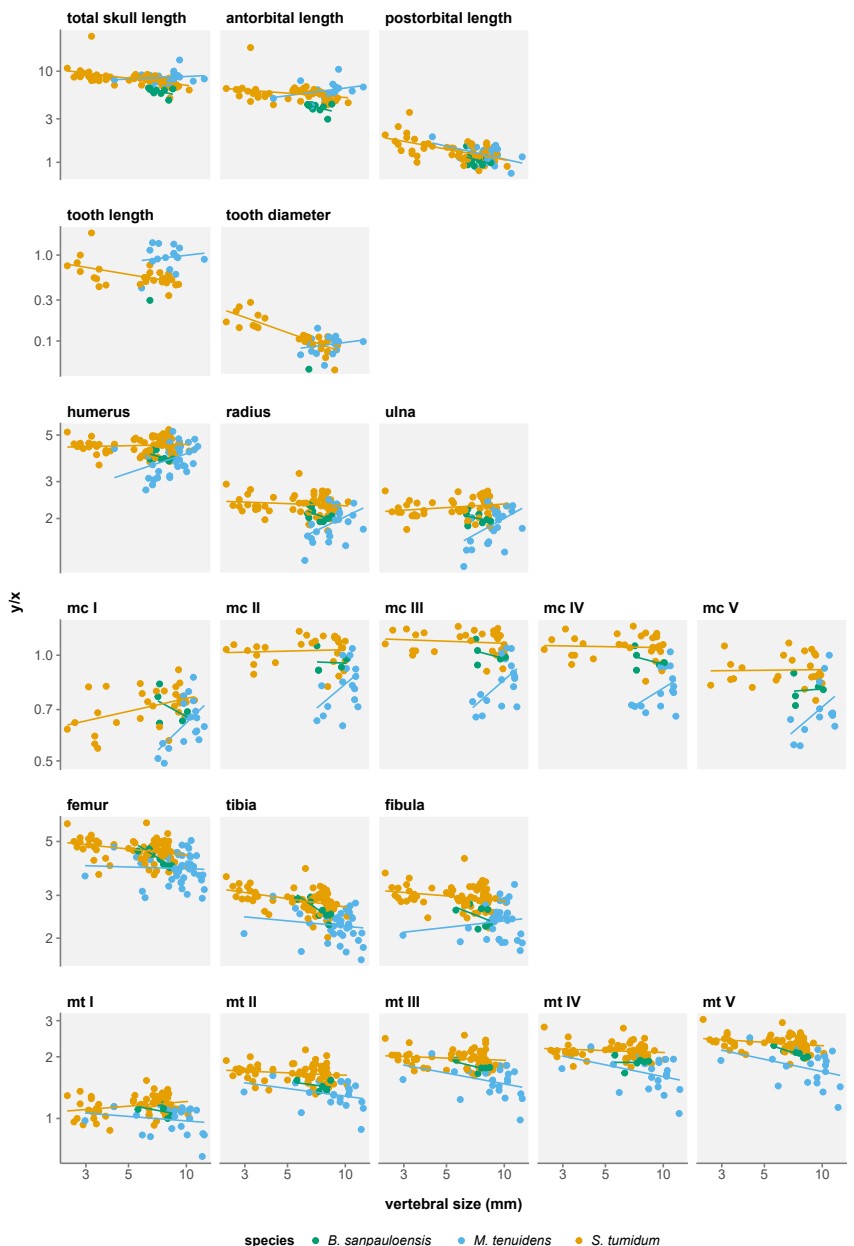

**Figure 2 Measurement/size ratios in function of size under the three-species hypothesis.** Logged scale.

In "*Stereosternum*", metacarpal I and metatarsal I also display a significant positive allometry, but femur, tibia, fibula and metacarpal V show a significant negative allometry (Fig. 2). Uniquely in mesosaur "species", the posterior autopod also shows a slight significant negative allometry. This reflects a reduction in limb size with respect to the body. "*Stereosternum*" being the best-documented taxon here, these results are more representative than for the two other species.

**Table 2** Allometry coefficients of logged tooth and skull measurements with respect to total skull length.

| | Allometry coefficient $\beta$ | SE | $t$-value | $p$-value |
|---|---|---|---|---|
| *"B. sanpauloensis"* | | | | |
| Tooth length | – | – | – | – |
| Tooth diameter | – | – | – | – |
| Skull antorbital length | 0.134 | 0.086 | 1.563 | 0.157 |
| Skull postorbital length | −0.925 | 0.522 | −1.771 | 0.114 |
| *M. tenuidens* | | | | |
| Tooth length | −0.051 | 0.276 | −0.186 | 0.855 |
| Tooth diameter | −0.103 | 0.260 | −0.398 | 0.696 |
| Skull antorbital length | 0.187 | 0.058 | 3.206 | **0.005** |
| Skull postorbital length | −0.590 | 0.156 | −3.785 | **0.001** |
| *"S. tumidum"* | | | | |
| Tooth length | −0.091 | 0.097 | −0.936 | 0.357 |
| Tooth diameter | −0.678 | 0.107 | −6.336 | **<0.001** |
| Skull antorbital length | 0.121 | 0.021 | 5.761 | **<0.001** |
| Skull postorbital length | −0.172 | 0.076 | −2.249 | **0.029** |
| Single-species hypothesis | | | | |
| Tooth length | 0.282 | 0.115 | 2.456 | **0.018** |
| Tooth diameter | −0.558 | 0.092 | −6.082 | **<0.001** |
| Skull antorbital length | 0.147 | 0.017 | 8.506 | **<0.001** |
| Skull postorbital length | −0.315 | 0.059 | −5.314 | **<0.001** |

**Notes.**
Significant values ($p < 0.05$) are indicated in bold.

### ANOVAs

We measure a significant effect of species on allometry coefficients for eight measurements: total skull length, antorbital length, postorbital length, tooth length, humerus, femur, tibia, and fibula length (Table 4). However, rather than indicating a real difference between the species, this likely reflects a sampling bias. In the near absence of *Mesosaurus* and *"Brazilosaurus"* juveniles in our dataset, values for these species form clusters concentrated in the adult–size region, which distorts linear regressions (Fig. 2). Therefore, these regressions do not fully account for the ontogenetic trajectories of the measurements.

## Single-species hypothesis

In a recent study, *Piñeiro et al. (2021)* proposed *Mesosaurus tenuidens* to be the only valid species within the clade by statistically rejecting most characters used to discriminate between the three previously recognized mesosaur species. To account for this new hypothesis, we reproduced our analysis with all specimens (re)assigned to *Mesosaurus*.

### Skull

All three skull measurements show negative allometry when compared to body size, but the linear relationship is only significant for postorbital length and total skull length (Table 1). When measuring allometry with respect to total skull length, the signal is unambiguous,

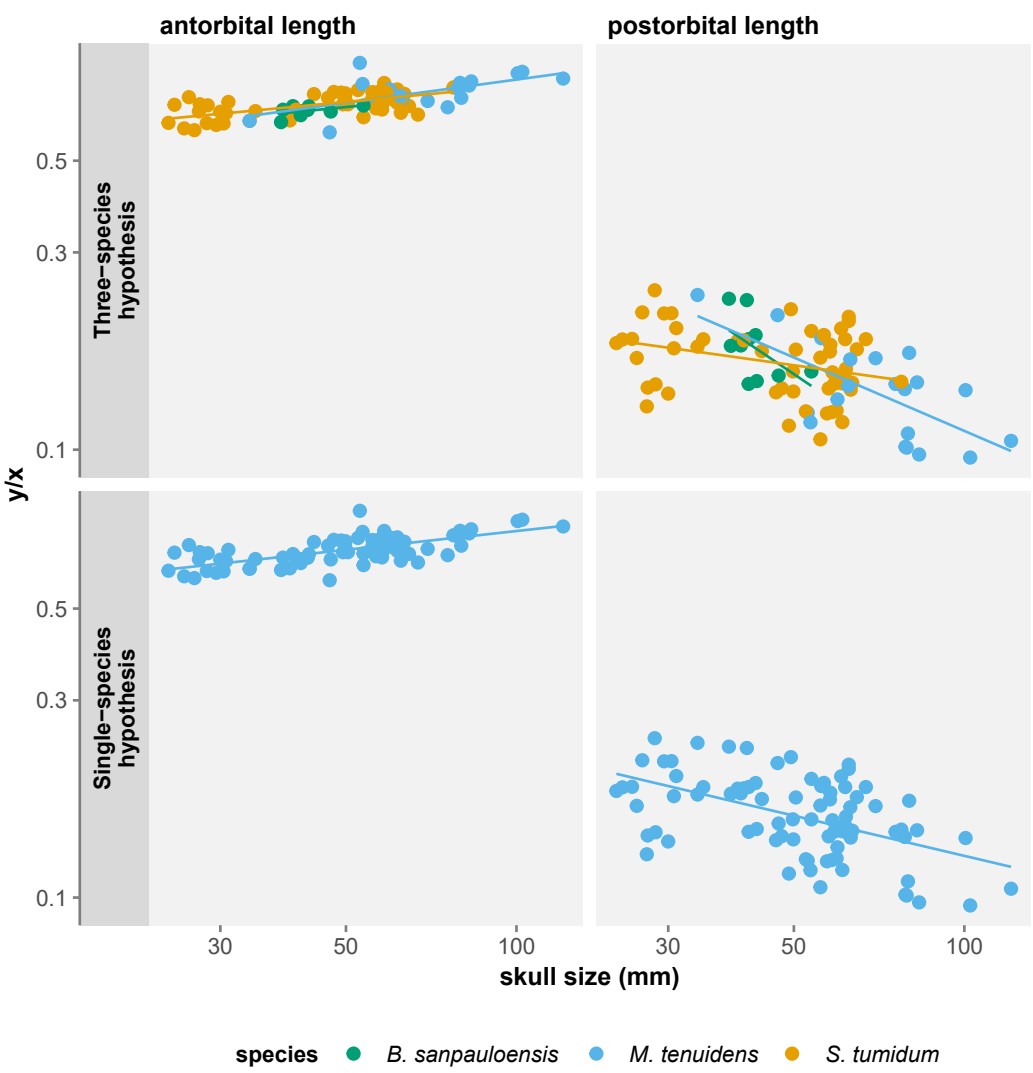

**Figure 3** Comparison of skull measurements/total skull length ratios in function of total skull length under the three-species and the single-species hypothesis. Logged scale.

with antorbital length showing positive allometry and postorbital length negative allometry (Fig. 3). Similar to the three-species hypothesis, our results with one species point to an elongation of the snout elongates and a reduction of postorbital length, while overall skull length proportionally decreases with respect to the body.

These results differ from those of *Núñez Demarco, Ferigolo & Piñeiro (2022)* who measured an isometric relationship of antorbital and postorbital length to total skull length. However, this can be easily explained by a difference in the statistical power of our datasets. We use a much larger dataset (84 specimens *vs.* 19), including more juveniles and larger specimens, which refines the results by producing smaller confidence intervals. It is therefore likely that the isometry in skull proportion measured by *Núñez Demarco, Ferigolo & Piñeiro (2022)* is a statistical artifact caused by the size of their dataset.
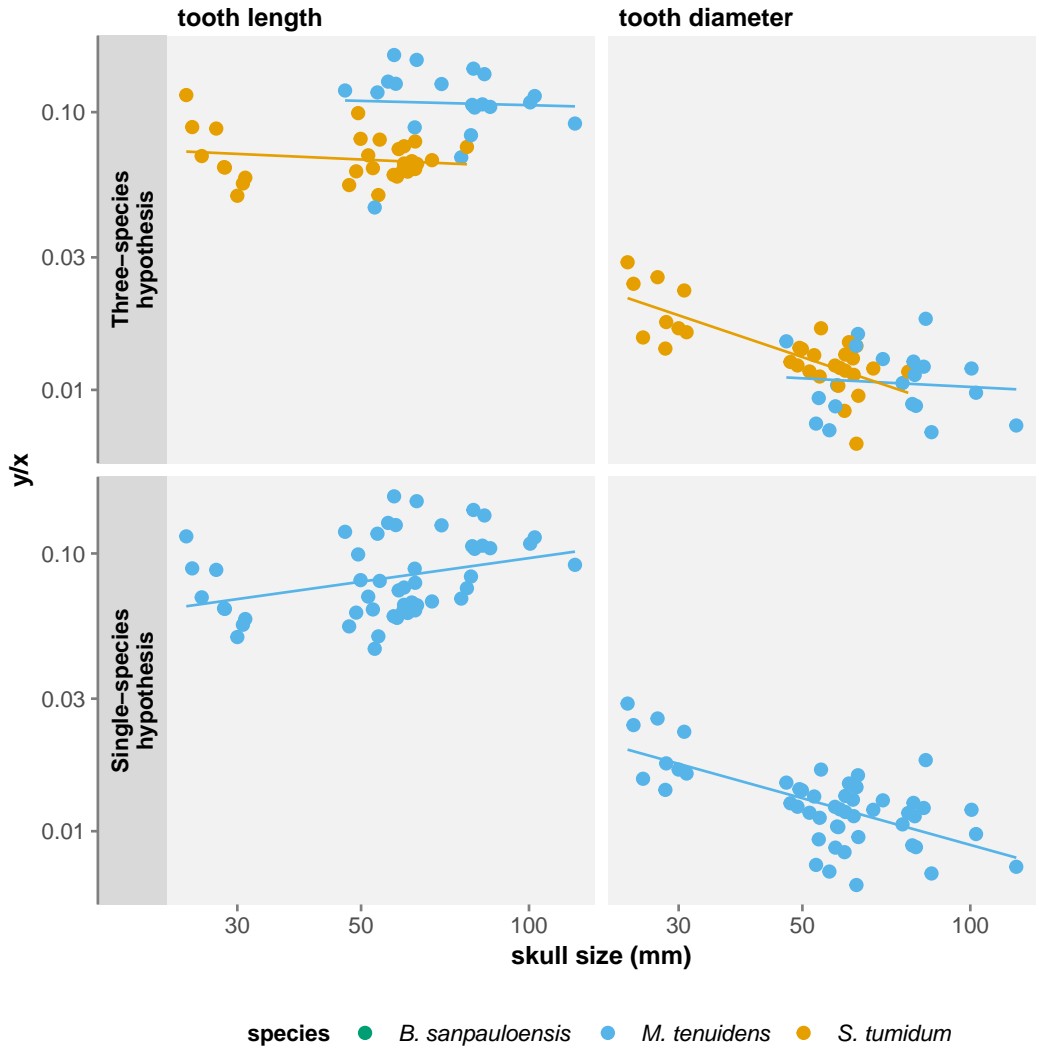

**Figure 4  Comparison of tooth measurements/total skull length ratios in function of total skull length under the three-species and the single-species hypothesis.** Logged scale.

## Teeth

Under the single-species hypothesis, tooth length shows positive but non-significant allometry whereas diameter shows a significant negative allometry with respect to body size. However, when compared to skull size, tooth length exhibits a strong significant positive allometry and tooth diameter a significant negative allometry (Table 2). In practice, tooth diameter remains the same in adults as in juveniles, but teeth elongate throughout ontogeny and become more needle-like.

## Limbs

There is a strong trend to hind limb reduction when considering a single mesosaur taxon (Fig. 5). All eight long bone measurements display negative allometry coefficients in relation to body size and this relationship is significant for seven of them (Table 3D). Only

**Table 3 Allometry coefficient of logged size ratios for long bone lengths.** A. "*B. sanpauloensis*". B. *M. tenuidens*. C. "*S. tumidum*". D. *Single-species hypothesis.*

| A. "*B. sanpauloensis*" | Allometry coefficient $\beta$ | SE | *t*-value | *p*-value |
|---|---|---|---|---|
| Humerus | −0.044 | 0.033 | −1.321 | 0.189 |
| Radius | −0.081 | 0.054 | −1.497 | 0.137 |
| Ulna | −0.083 | 0.057 | −1.449 | 0.151 |
| mc I | 0.062 | 0.058 | 1.078 | 0.285 |
| mc II | −0.122 | 0.062 | −1.988 | 0.052 |
| mc III | −0.159 | 0.063 | −2.509 | 0.015 |
| mc IV | −0.173 | 0.063 | −2.728 | 0.009 |
| mc V | −0.146 | 0.081 | −1.809 | 0.077 |
| Anterior autopod | 0.050 | 0.212 | 0.236 | 0.852 |
| Femur | −0.154 | 0.030 | −5.078 | **<0.001** |
| Tibia | −0.212 | 0.035 | −6.119 | **<0.001** |
| Fibula | −0.167 | 0.039 | −4.268 | **<0.001** |
| mt I | −0.026 | 0.036 | −0.709 | 0.480 |
| mt II | −0.142 | 0.035 | −4.068 | **<0.001** |
| mt III | −0.163 | 0.040 | −4.108 | **<0.001** |
| mt IV | −0.157 | 0.039 | −4.042 | **<0.001** |
| mt V | −0.185 | 0.040 | −4.630 | **<0.001** |
| Posterior autopod | – | – | – | – |

| B. *M. tenuidens* | Allometry coefficient $\beta$ | SE | *t*-value | *p*-value |
|---|---|---|---|---|
| Humerus | 0.308 | 0.125 | 2.473 | **0.018** |
| Radius | 0.445 | 0.171 | 2.599 | **0.016** |
| Ulna | 0.515 | 0.199 | 2.588 | **0.016** |
| mc I | 0.646 | 0.244 | 2.650 | **0.017** |
| mc II | 0.569 | 0.282 | 2.018 | 0.062 |
| mc III | 0.590 | 0.218 | 2.713 | **0.016** |
| mc IV | 0.417 | 0.217 | 1.923 | 0.077 |
| mc V | 0.573 | 0.308 | 1.859 | 0.090 |
| Anterior autopod | 0.527 | 0.607 | 0.867 | 0.450 |
| Femur | −0.024 | 0.077 | −0.31 | 0.758 |
| Tibia | −0.074 | 0.077 | −0.952 | 0.347 |
| Fibula | 0.089 | 0.112 | 0.799 | 0.431 |
| mt I | −0.073 | 0.09 | −0.812 | 0.425 |
| mt II | −0.124 | 0.082 | −1.524 | 0.143 |
| mt III | −0.175 | 0.096 | −1.828 | 0.082 |
| mt IV | −0.192 | 0.105 | −1.816 | 0.084 |
| mt V | −0.199 | 0.115 | −1.728 | 0.101 |

**Table 3** (*continued*)

| B. *M. tenuidens* | Allometry coefficient β | SE | *t*-value | *p*-value |
|---|---|---|---|---|
| Posterior autopod | −0.122 | 0.053 | −2.293 | 0.262 |

| C. "*S. tumidum*" | Allometry coefficient β | SE | *t*-value | *p*-value |
|---|---|---|---|---|
| Humerus | 0.021 | 0.027 | 0.771 | 0.444 |
| Radius | −0.034 | 0.036 | −0.936 | 0.353 |
| Ulna | 0.057 | 0.034 | 1.698 | 0.095 |
| mc I | 0.150 | 0.056 | 2.683 | **0.012** |
| mc II | 0.017 | 0.045 | 0.366 | 0.717 |
| mc III | −0.024 | 0.039 | −0.616 | 0.543 |
| mc IV | −0.012 | 0.041 | −0.279 | 0.782 |
| mc V | 0.007 | 0.058 | 0.124 | 0.902 |
| Anterior autopod | −0.025 | 0.066 | −0.370 | 0.716 |
| Femur | −0.081 | 0.029 | −2.779 | **0.007** |
| Tibia | −0.113 | 0.03 | −3.773 | **<0.001** |
| Fibula | −0.065 | 0.034 | −1.875 | 0.065 |
| mt I | 0.074 | 0.034 | 2.173 | **0.033** |
| mt II | −0.038 | 0.029 | −1.304 | 0.197 |
| mt III | −0.035 | 0.031 | −1.114 | 0.269 |
| mt IV | −0.031 | 0.028 | −1.089 | 0.280 |
| mt V | −0.056 | 0.027 | −2.069 | **0.043** |
| Posterior autopod | −0.140 | 0.061 | −2.310 | **0.028** |

| D. Single–species hypothesis | Allometry coefficient β | SE | *t*-value | *p*-value |
|---|---|---|---|---|
| Humerus | −0.041 | 0.035 | −1.184 | 0.239 |
| Radius | −0.109 | 0.043 | −2.502 | **0.014** |
| Ulna | −0.044 | 0.046 | −0.949 | 0.346 |
| mc I | 0.061 | 0.057 | 1.074 | 0.288 |
| mc II | −0.113 | 0.062 | −1.831 | 0.073 |
| mc III | −0.154 | 0.064 | −2.411 | **0.020** |
| mc IV | −0.157 | 0.062 | −2.522 | **0.015** |
| mc V | −0.130 | 0.082 | −1.589 | 0.120 |
| Anterior autopod | −0.120 | 0093 | −1.281 | 0.212 |
| Femur | −0.153 | 0.029 | −5.303 | **<0.001** |
| Tibia | −0.215 | 0.031 | −6.875 | **<0.001** |
| Fibula | −0.170 | 0.037 | −4.600 | **<0.001** |
| mt I | −0.037 | 0.035 | −1.060 | 0.292 |
| mt II | −0.150 | 0.034 | −4.470 | **<0.001** |
| mt III | −0.166 | 0.038 | −4.405 | **<0.001** |
| mt IV | −0.161 | 0.038 | −4.247 | **<0.001** |
| mt V | −0.187 | 0.039 | −4.774 | **<0.001** |
| Posterior autopod | −0.152 | 0.053 | −2.863 | **0.007** |

**Table 4   ANOVA results for the effect of species on measurements.**

|  | df | F | p-value |
|---|---|---|---|
| Total skull length | 2 | 13.087 | **0.000** |
| Antorbital length | 2 | 11.774 | **0.000** |
| Postorbital length | 2 | 4.576 | **0.013** |
| Tooth length | 1 | 59.118 | **0.000** |
| Tooth diameter | 1 | 2.922 | 0.094 |
| Humerus | 2 | 3.362 | **0.038** |
| Radius | 2 | 1.374 | 0.258 |
| Ulna | 2 | 1.923 | 0.152 |
| mc I | 2 | 2.620 | 0.082 |
| mc II | 2 | 0.926 | 0.403 |
| mc III | 2 | 0.445 | 0.644 |
| mc IV | 2 | 0.380 | 0.686 |
| mc V | 2 | 0.088 | 0.916 |
| Femur | 2 | 6.282 | **0.002** |
| Tibia | 2 | 3.827 | **0.025** |
| Fibula | 2 | 4.507 | **0.013** |
| mt I | 2 | 2.912 | 0.059 |
| mt II | 2 | 1.575 | 0.213 |
| mt III | 2 | 1.058 | 0.351 |
| mt IV | 2 | 0.792 | 0.456 |
| mt V | 2 | 0.501 | 0.608 |

**Notes.**

Significant values ($p < 0.05$) are indicated in bold.

metatarsal I length is not significantly isometric in growth. Posterior autopod length also shows a significant negative allometry.

In contrast, forelimb long bones do not show such a reduction in size. Only metacarpals display significant negative allometry, with the exception of metacarpal I that has a non-significant positive allometry coefficient. Thus, while stylopod and zeugopod do not vary in proportions, it appears the manus is reduced during ontogeny.

At first glance, it could seem that these results do not match those of *Núñez Demarco, Ferigolo & Piñeiro (2022)*, who report isometric growth in mesosaur limbs. Nevertheless, our findings are compatible with theirs. What *Núñez Demarco, Ferigolo & Piñeiro (2022)* measure is that long bones grow isometrically with respect to their respective stylopod, humerus length for fore limbs and femur for hind limbs. What we measure is the allometry of these bones with respect to size. Here we show that femur length displays a significant negative allometry with respect to size, and so do most other limb bones. In sum, the isometric growth measured by *Núñez Demarco, Ferigolo & Piñeiro (2022)*, and by extension their other results for limbs, is likely an effect of using different reference frames.

## Environmental distribution

Figure 6 shows the distribution of size classes in the main facies of the Irati-Whitehill Sea. In our sample, small mesosaurs (*i.e.,* juveniles) are overrepresented in limestone, although

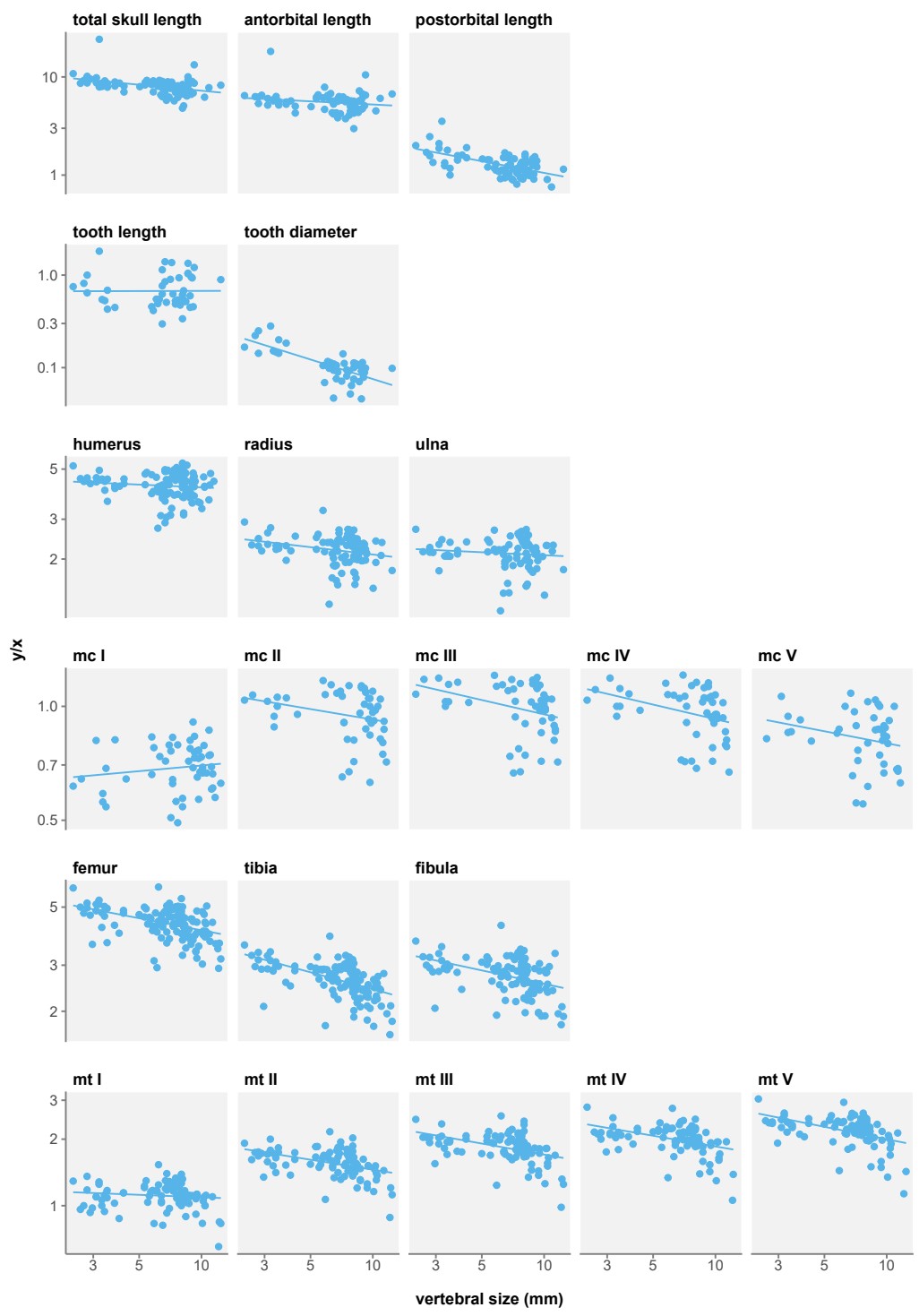

**Figure 5   Measurement/size ratios in function of size under the single-species hypothesis.** Logged scale.

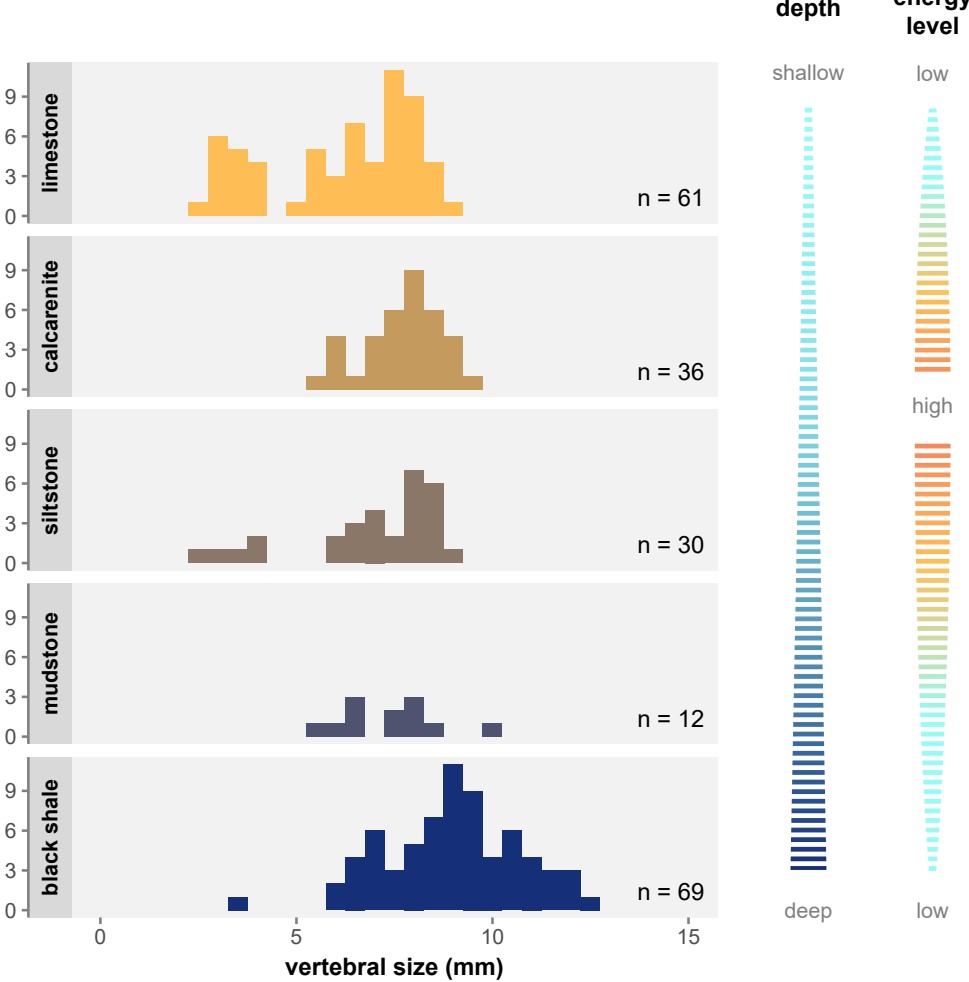

**Figure 6** **Size distribution of vertebral size per geological facies under the single-species hypothesis.**
Facies are organized top to bottom along a gradient of depth as depicted to the right, and following the deposition models from *Xavier et al. (2018)* and *Ng, Vega & Maranhão (2019)*. The energy level gradient on the right corresponds to grain size in the sediment. There is a clear overrepresentation of juveniles in limestone, as well as an overrepresentation of large specimens in black shale.

they also occur in siltstone and black shales. Mid-sized mesosaurs are well represented in all facies despite frequency variations due to sample size differences. Contrary to what has been hypothesized (*Núñez Demarco et al., 2018*), we do not find that large specimens are only found in the marginal regions of the basin. Instead, in our dataset, very large specimens (vertebral length ≥ 10 mm) are exclusively found in mudstone and black shales, facies that correspond to deeper, more pelagic, and more anoxic environments (*Xavier et al., 2018*; *Bastos et al., 2021*).

## DISCUSSION

### Missing juveniles and single-species hypothesis

One of the most significant results of our study is not found with regards to size and proportional changes in mesosaur anatomy but instead in the distribution of body size itself. Despite the 270 specimens examined here, there are a number of surprising gaps in the size range covered by our sample (Fig. 7). First, the few "*Brazilosaurus*" specimens studied here are concentrated in the adult size range of "*Stereosternum*", but no juveniles or subadults are preserved. Second, there is a clear divide between the youngest "*Stereosternum*" specimens on the one hand and subadults and adults of that species on the other hand. Third, and more importantly, only two specimens of *Mesosaurus* are the size of juvenile "*Stereosternum*", whereas all other specimens range from the size of "*Stereosternum*" subadults to larger sizes than the largest "*Stereosternum*" specimens. These gaps in size distribution have a direct effect on the interpretation of allometric measurements as they alter the results of linear regressions, making them less representative of ontogenetic changes.

*Piñeiro et al. (2021)* highlighted that most features employed to define mesosaur species are not statistically supported or are practically unusable for identification. Characters like the shape of the interclavicle, tooth length, or presacral vertebral count are highly preservation-dependent, whereas others such as the degree of pachyostosis of ribs or the skull-neck length ratio are subjective and ontogenetically variable (for pachyostosis, see also *Klein et al., 2019*). This lack of reliable characters led the authors to suggest that both "*Stereosternum tumidum*" and "*Brazilosaurus sanpauloensis*" are junior synonyms of *Mesosaurus tenuidens*.

The ontogenetic variability of supposedly taxonomic characters led to an age partitioning of mesosaur specimens into the different "species". On the one hand, juvenile and subadult specimens tended to be identified as "*Stereosternum*", because they bear shorter conical teeth and are usually preserved complete, displaying the maximum number of presacral vertebrae. Both features were formally admitted as prominent characters for mesosaur identification.

On the other hand, the largest mesosaur specimens bear longer, thinner teeth and have more markedly pachyostotic ribs, and they tended historically to be attributed to *Mesosaurus*. As for "*Brazilosaurus*" specimens, they were usually identified on the basis of their cervical vertebral count, which is however a misidentification as the result of a displaced rib cage (A. Verrière, 2022, pers. obs.). In fact, even the original description of the species (*Shikama & Ozaki, 1966*) miscounted the number of cervicals due to the posterior displacement of dorsal ribs (A. Verrière, 2022, pers. obs.), which led to the erroneous erection of this feature as a defining character. To summarize, there are several reasons as to why mesosaur invalidated "species" occupy different regions of the size range.

Rather than a sampling bias, the size distribution gaps in our dataset more likely reflect a species-identification bias. Our data shows consequent overlap in the morphometrics of all three species. "*Brazilosaurus*" is virtually indistinguishable from "*Stereosternum*", whereas the old definition of *Mesosaurus* seems to have mostly representatives on the higher side of the size spectrum (Fig. 7). Furthermore, the low significance of allometry coefficients

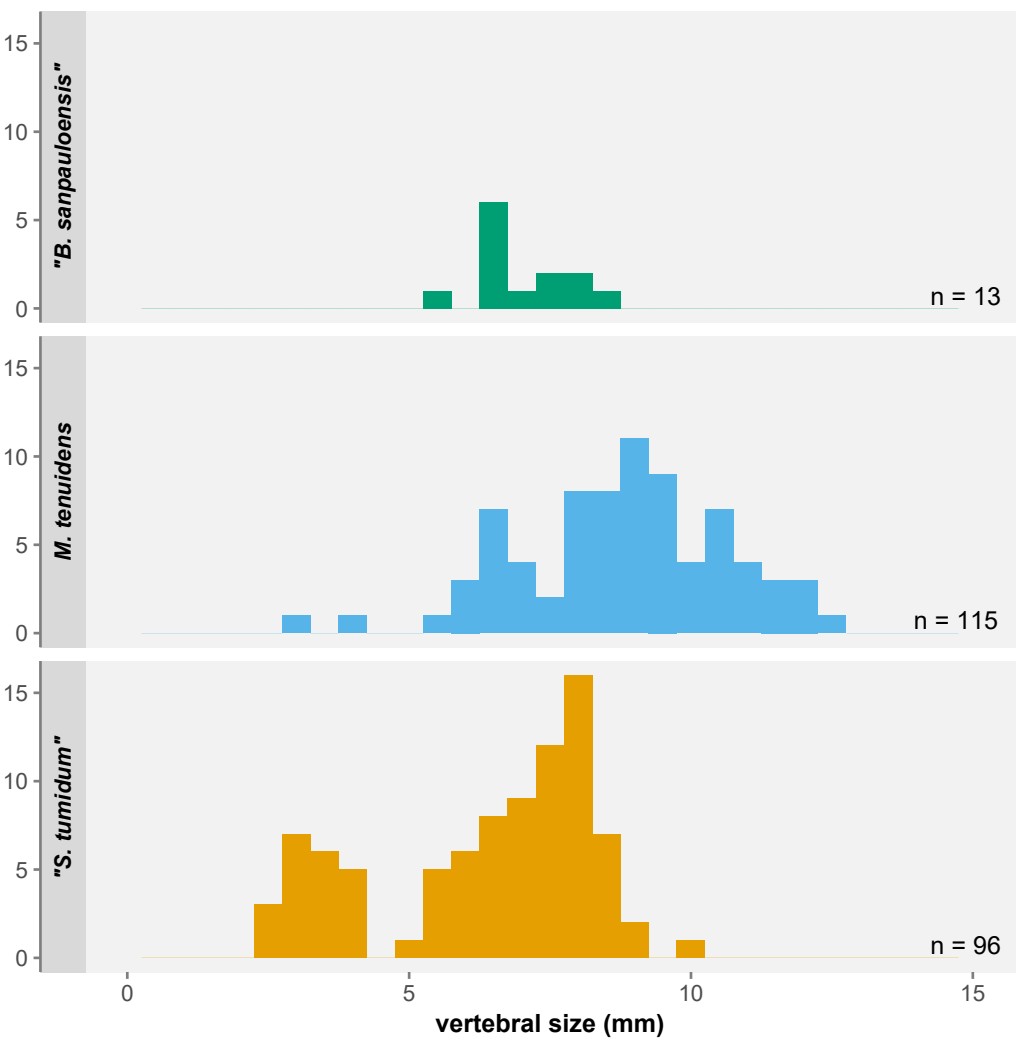

**Figure 7  Histogram of average vertebral size under the three-species hypothesis.** Small/juvenile specimens are notably overrepresented in the "*S. tumidum*" group, whereas large individuals are only represented in the *M. tenuidens* group.

(Tables 1–3) is also likely an effect of the small number of juveniles of the *Mesosaurus* type in our dataset, although more juvenile specimens have been described that we could not include (*Piñeiro et al., 2012a*).

Our results therefore corroborates the single-species hypothesis of *Piñeiro et al. (2021)* and their discarding of the former characters as ontogenetically variable. In fact, the three previously identified "species" more probably represented three artificial types: the "*Brazilosaurus*"-type is a poorly preserved mesosaur with displaced ribs, the "*Stereosternum*"-type is a size class encompassing juveniles to young adults, and the "*Mesosaurus*"-type represents adults with more extreme sizes and morphologies. Consequently, hereafter we will only consider our results in the light of the single-species hypothesis.

## Environmental partitioning

Even when considering *Mesosaurus* as the only valid mesosaur taxon, one puzzling difference between the previously distinct types cannot be explained as an effect of ontogenetic changes or dubious characters alone. Some authors pointed out that the three types were not equally distributed across the facies of the Irati-Whitehill Sea. *Oelofsen & Araújo (1983)* and *Rossmann (2002)* noted that "*Brazilosaurus*" and "*Stereosternum*" were more commonly found in limestone deposits, whereas *Mesosaurus* was more frequent in oily black shales. Since these publications however, mesosaur specimens have been described in other facies and the former species have been synonymized, meaning their observations must be considered with caution.

Nevertheless, our study demonstrate that size classes are not homogeneously distributed across lithological facies: juveniles are overrepresented in limestone, and very large/old specimens in black shales (Fig. 6). As argued above, flaws in the formerly admitted diagnostic characters have led to juveniles and young adults being preferentially assigned to "*Stereosternum*" and old adults to *Mesosaurus* (Fig. 7). Because we look at articulated specimens, it is also fair to assume that the specimens were buried relatively close to their life environment. Therefore, what *Oelofsen & Araújo (1983)* and *Rossmann (2002)* noticed were likely the signs of ontogenetic stage partitioning rather than species partitioning.

The overrepresentation of juveniles and large adult in some facies suggests that mesosaurs preferentially occupied some environments at different stages of their life. Juveniles were mostly present in the shallower, more coastal sections of the carbonate ramp, while medium-sized mesosaurs were present in all areas, and the largest and oldest individuals were restricted to pelagic and anoxic regions (Fig. 6). Thus, if mesosaurs were not confined to a specific region of their environment, it seems that, overall, they gradually moved away from the coast during their growth.

The pelagic migration of mesosaurs with growth raises the question of their reproduction. Associated articulated juveniles and adults, even gravid individuals, have been found in limestone, siltstone, and mudstone (*Piñeiro et al., 2012a*; *Bickelmann & Tsuji, 2018*), but the greater frequency of juveniles in limestone suggests that mesosaurs were mostly born near the coast. If they were oviparous like most reptiles, this would have been easily explained by the necessity to lay eggs on land. However, evidence suggests instead that mesosaurs were viviparous, and possibly even cared for their young (*Piñeiro et al., 2012a*). Mesosaurs could therefore have given birth anywhere in the sea, making the causes of a partitioning less clear. To try to explain this, we must look at our morphometric analyses.

## Ontogenetic changes and diet shifts

Our results show that mesosaur growth is marked by several morphological changes: an elongation of the teeth relative to the skull (Fig. 3), an elongation of the snout (Fig. 4), and a reduction of limbs (Fig. 5). Limb reduction throughout ontogeny hints at a difference in locomotory habits between juveniles and adults. Mesosaurs are swimming reptiles, and the degree of involvement of the limbs in locomotion is still debated. Some authors support the idea of mainly tail-driven mesosaurs (*Braun & Reif, 1985*; *Villamil et al., 2016*) while others suggested a potentially larger role of limbs in propulsion (*Da Silva &*

*Sedor, 2017*; *MacDougall et al., 2020*). Yet, the ontogenetic limb reduction we expose seems incompatible with the conception that limbs were crucial for locomotion. Instead, our results provide evidence for a decreasing involvement of limbs in mesosaur swimming and weigh in favor of a tail-driven model of mesosaur swimming throughout their ontogeny.

The association of limb reduction with cranial and dental modifications strongly suggest the existence of locomotion and dietary shift in mesosaur. Such shifts are not rare in reptiles but are usually associated with dietary partitioning between juveniles and adults, meaning they exploit different food sources at different stages of their growth (*Arthur, Boyle & Limpus, 2008*; *Gignac & Erickson, 2015*; *Dick, Schweigert & Maxwell, 2016*; *Wang et al., 2017*). In the case of mesosaurs, a change in diet could explain the environmental migration we observe in size classes (Fig. 6). As mesosaurs grew, they would have shifted from an abundant food source in the coastal environment to another one more prevalent in pelagic regions or other water layers. Preying on different food sources in different water layers would have been a way to avoid competition between ontogenetic stages in a sea where resources were scarce. Different feeding habits would also explain the presence of juveniles in pelagic environments, as young mesosaurs could have been born anywhere in the sea but would have preferentially migrated to the coastal region looking for food. Yet, there remains the question of the alternative food source.

Traditionally, mesosaurs are considered filter-feeders that preyed on pygocephalomorph crustaceans, based on stratigraphic occurrences (*Oelofsen & Araújo, 1983*; *Piñeiro et al., 2012b*; *Xavier et al., 2018*; *Ng, Vega & Maranhão, 2019*) as well as on gastric contents and coprolites (*Piñeiro et al., 2012b*; *Ramos, 2015*; *Silva et al., 2017*). While this applies to adults, it is unclear what the diet of juveniles could have been. Mesosaur-bearing strata of the Irati-Whitehill Sea deposits are rich in mesosaurs and pygocephalomorphs remains but scarce in other types of fossils. Only rare invertebrate traces, bivalves, and paleonisciform fish scales have been described (*Oelofsen & Araújo, 1983*; *Piñeiro et al., 2012b*; *Matos et al., 2017*; *Xavier et al., 2018*), although recent works have revised the abundance of paleonisciform fishes upwards (*Xavier et al., 2018*). This raises a puzzling question: if juvenile mesosaurs did not have the same diet as adults, what could they possibly have been eating?

Our morphometric analysis shows that young mesosaurs would have had longer limbs, shorter snouts and shorter teeth than adults, a morphology better suited to active hunting than that of adults. Among the known Irati-Whitehill fauna, only pygocephalomorph crustaceans and paleonisciform fishes could have been the subject of active hunting. Thus, juvenile mesosaurs could have actively preyed on small fish or large pygocephalomorphs, while mesosaurs would have slowly filtered waters to catch smaller pygocephalomorphs or even larvae. The coastal environment would have been better oxygenated and richer in nutrients, fostering the growth of larger preys, whereas smaller crustaceans could have occupied the harsher pelagic waters. Unfortunately, the ecology of pygocephalomorph crustaceans or paleonisciform fish is very poorly known. One way of testing this hypothesis would be to examine size distribution across facies for fish and crustaceans and see if there is a trend, similar to what we did for mesosaurs.

Another possible answer is that juvenile mesosaurs fed on animals that did not preserve well under the conditions where mesosaurs were buried. High energy conditions in the

mid-coastal environments as well as acidic anoxic waters in the black shale environment might have hampered the preservation of some animals. Moreover, young mesosaurs could have eaten jellyfish or soft invertebrates with little to no mineralized tissues, which greatly reduces the chances at fossilizing the latter. Although the sea's hypersaline conditions would not have favored such animals (*Piñeiro et al., 2012b*; *Bastos et al., 2021*), there are occurrences of soft invertebrates in hypersaline contexts (*Parker, 1959*; *Tanner, Glenn & Moore, 1999*; *Gilabert, 2001*). Yet, there are many cases of outstanding preservation in the Irati-Whitehill sediments producing complete articulated fossils (*Modesto, 1999*; *Modesto, 2010*; *Piñeiro et al., 2012b*; *Núñez Demarco et al., 2018*), sometimes even showing soft tissue preservation (*Piñeiro et al., 2012b*; *Piñeiro et al., 2012a*; *MacDougall et al., 2020*). With such conditions of preservation, it seems very unlikely that no trace of other invertebrates would remain. It is possible, however, that soft animal fossils are yet to be discovered, and that further study of the mesosaur-bearing Irati-Whitehill deposits will reveal them.

## CONCLUSIONS

After examining a large sample of mesosaur morphometric measurements, we present evidence of ontogenetic changes in these animals as well as insights into their taxonomy. First, our results demonstrate that the three previously identified species occupy incomplete portions of the size range of mesosaurs and show very minor differences in body proportions otherwise. This supports the proposition of *Piñeiro et al. (2021)* that *Mesosaurus tenuidens* is the only valid species and indicates that "*Stereosternum tumidum*" and "*Brazilosaurus sanpauloensis*" likely represent immature stages or incomplete specimens of *Mesosaurus*.

Our results also highlight the following ontogenetic trends:

- a reduction in skull length accompanied by an elongation of the snout within the skull
- an elongation of teeth
- a reduction in hind limb length
- a reduction in manus length.

These ontogenetic trends are associated with a partitioning of ontogenetic stages across facies following a gradient of depth and distance to the coast. This suggests that mesosaurs underwent a diet and lifestyle transition during their growth: juveniles had an active predatory lifestyle in a coastal environment, while old adults adopted a more filter-feeding diet in more pelagic and anoxic waters. Our hypothesis is that mesosaurs exploited different food sources during their lifetime, but further study of the Irati-Whitehill Sea fauna will be necessary to truly test this.

### Institution Acronyms

| | |
|---|---|
| **AMNH** | American Museum of Natural History, New York, NY, USA. |
| **BSPG** | Bayerische Staatssammlung für Paläontologie und Geologie, Munich, Germany. |
| **CM** | Carnegie Museum of Natural History, Pittsburgh, PA, USA. |
| **IGC** | Instituto Geográfico e Cartográfico do Estado de São Paulo, São Paulo, SP, Brazil. |

| GPIT | Geologisch-Palaeontologisches Institut Tübingen, Tübingen, Germany. |
| MB | Museum für Naturkunde, Berlin, Germany. |
| MCZ | Museum of Comparative Zoology, Cambridge, MA, USA. |
| MfNM | Museum für Naturkunde Magdeburg, Magdeburg, Germany. |
| MNHN | Muséum National d'Histoire Naturelle, Paris, France. |
| MZSP | Museu de Zoologia da USP, São Paulo, SP, Brazil. |
| NMNS | National Museum of Nature and Science, Tokyo, Japan. |
| NHMW | Naturhistorisches Museum Wien, Vienna, Austria. |
| NRM | Naturhistoriska Riksmuseet, Stockholm, Sweden. |
| PIMUZ | Paläontologische und Zoologische Museum, Zürich, Switzerland. |
| ROM | Royal Ontario Museum, Toronto, ON, Canada. |
| SMF | Senckenberg Museum Frankfurt, Frankfurt am Main, Germany. |
| SMNK | Staatliches Museum für Naturkunde Karlsruhe, Karlsruhe, Germany. |
| SMNS | Staatliches Museum für Naturkunde Stuttgart, Stuttgart, Germany. |
| UFRGS | Universidade Federal do Rio Grande do Sul, Porto Alegre, RS, Brazil. |
| UFSCar | Universidade Federal de São Carlos, São Carlos, SP, Brazil. |
| URCR | Universidade Estadual Paulista, Campus de Rio Claro, Rio Claro, SP, Brazil. |
| USNM | National Museum of Natural History, Washington, DC, USA. |

## ACKNOWLEDGEMENTS

For granting us access to specimens and for their help and hospitality during our collection visits, we are extremely thankful to Carl M. Mehling (AMNH), Oliver Rauhut (BSPG), Juliana Leme (IGC), Ingmar Werneburg (GPIT), Thomas Schossleitner (MB), Jessica Cundiff (MCZ), Michael Buchwitz (MfNM), Nour-Eddine Jalil (MNHN), Alberto B. Carvalho(MZSP), Ursula Göhlich (NHMW), Thomas Mörs (NRM), Christian Klug (PIMUZ), Kevin Seymour (ROM), Rainer Brocke (SMF), Eberhard Frey (SMNK), Rainer Schoch and Erin Maxwell (SMNS), Cesar Leandro Schulz, Thiago Carlisbino, and Voltaire Paes Neto (UFRGS), Aline Ghilardi, Tito Aureliano, and Marcelo A. Fernandes (UFSCar), Reinaldo Bertini (URCR), and Amanda Millhouse (USNM). Many thanks to Amy Henrici (CM) and Makoto Manabe (NMNS) for providing us with high resolution photographs of the mesosaur specimens in their collections. Thanks to Yara Haridy for her ideas and support that helped spark this project. We would like to thank Esther Ullrich-Lüter for catalyzing the drafting of this manuscript. We are also very thankful to Vanuhi Hambardzumyan and Mark MacDougall for providing very helpful comments at different stages of the manuscript. A. Verrière would like to address very special thanks to Heitor Satorelli for his friendship, for his heartfelt hospitality and for helping him navigate through Brazil. We would also like to thank reviewers Cesar L. Schulz, Graciela Piñeiro, and Flavio Pretto for their comments that greatly improved this manuscript.

### Funding

This work was supported by the German Research Foundation (DFG FR 2457/6-1). The funders had no role in study design, data collection and analysis, decision to publish, or preparation of the manuscript.

### Grant Disclosures

The following grant information was disclosed by the authors:
German Research Foundation: DFG FR 2457/6-1.

### Competing Interests

The authors declare there are no competing interests.

### Author Contributions

- Antoine Verrière conceived and designed the experiments, performed the experiments, analyzed the data, prepared figures and/or tables, authored or reviewed drafts of the article, collection visit, and approved the final draft.
- Jörg Fröbisch conceived and designed the experiments, authored or reviewed drafts of the article, collection visit, and approved the final draft.

### Data Availability

Raw data is available in the Supplemental Files.

### Supplemental Information

Supplemental information for this article can be found online at http://dx.doi.org/10.7717/peerj.13866#supplemental-information.

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
