# Peer review of "Ontogenetic, dietary, and environmental shifts in Mesosauridae"

_PeerJ, doi:10.7717/peerj.13866_

## Round 0.1 · original submission · Major Revisions

First of all, I need to apologize for the delay in making my decision, one promised review was waiting for too long, and finally, it did not come.

Anyway, your paper has two reviews that allow me to make this decision. One suggested point noted for both reviewers refers to the lack of discussion regarding the existence of adult and juvenile associations. In addition to this, our first reviewer stated many other points that require your full attention. Reading and discussing the recently published paper of Nuñez de Marco et al. (which is attached) would help you go deep in the explanations about the selection of the taxa and the methodology applied in this manuscript.

Consequently, the discussion section would also be more rigorous. I was hoping you could clarify the differences between your results and those in Nuñez de Marco et al., particularly in the isometric vs. allometric interpretations. I would appreciate it if you could consider all comments and include the modifications in your paper.

·

Basic reporting

This manuscript entitled Ontogenetic, dietary, and environmental shifts in Mesosauridae by Verrière and Fröbisch may be an interesting contribution to increase the knowledge of a group of basal amniotes that although being only represented in Gondwanan deposits, has recently attracted a lot of attention among researches around the world. Even though, the main goal of this manuscript is difficult to identify. On one side it appears to pursuit an attempt to test a recent taxonomic proposal that suggest that Mesosaurus tenuidens is the only supported mesosaur taxon and “Stereosternum tumidum” and “Brazilosaurus sanpauloensis” are their junior synonyms. On the other hand, the main subject seems to become more related to the exploration of the mesosaur ecology and biology that derive in a discussion based on the hypothetical existence of a mesosaur niche partitioning that is not consistent with the geological, sedimentological, taphonomic and biostratigraphic data already published and the proposal of a mesosaur diversity that is not supported by their own statistical results.

Experimental design

The research may be partially original -although several important problems were detected about ideas included in the discussion-, but also contains some development and purportedly conclusive states that are confluent with previous contributions on the area which were not enough discriminated respect to the corresponding authorship. I have warned the authors about this issue through several commentaries referred to certain paragraphs of the manuscript. Those commentaries can be seen in the annotated pdf attached to this report.
The methodology is striking too much similar to that used in a forthcoming paper by Nuñez Demarco et al., a paper that was recently accepted to be published in Acta Palaeontologica Polonica. Thus, it will be imperious that this paper that will be available online next week, be cited as a reference in the Verrière and Fröbisch manuscript (if it is finally accepted).
Despite the similarities of the applied methodology respect to the Núñez Demarco et al. paper, it has some inconvenient procedures that weaken the validation of their results. These aspects are as follow:
1- Under which criteria the different specimens used in the statistic analisis were classified as belonging to the three previously proposed species that were recently proved to be junior synonyms of Mesosaurus because of the lack of diagnostic characters that allow their identification? This is a key problem to solve because if the specimens were wrongly classified some of the obtained results could also be wrongly interpreted, invaliding all the posterior discussion and conclusions.
2- One must be very careful when defining allometries with beta values so close to zero (or alpha close to 1). If the error interval of the beta coefficient includes 0 (or if the 95% confidence interval of alfa includes 1), the result must be considered isometric, not allometric (see Rubenstein 1971; Leduc 1987; Anderson et al. 2016). As all your results are close to zero, the error intervals must be analyzed carefully to define statistically if they are not indicating isometry. This would possibly show that almost all your measurements (which coincidently are also the same used in Nuñez Demarco et al. 2022) are isometric as was demonstrated in this forthcoming paper.
3- Also coincident with the paper of Nuñez Demarco et al. you utilized the average length of the dorsal vertebrae in order to have a proxy for calculate the size of the analyzed specimens. This proxy was proposed by Nuñez Demarco et al (2018) who showed that it is valid and useful in mesosaurs, but especially if the average length of all the vertebrae present in the individual is considered, not only that of the dorsal ones. Curiously, this 2018 research by Nuñez Demarco et al. was not cited in the in review manuscript, but it should be in order to conduct this research in conformity with the ethical standars in science. Also, I recommend that authors explore the evolution of this criterion, which is utilized and improved in the Nuñez Demarco et al (2022) paper.

Validity of the findings

At its current structure, this manuscript does not show satisfactory robustness; particularly it needs changes in the methodology and the results should be consistent with the Discussion section, which includes hypotheses based on data that do not support them, as I will explain below (but see also the annotated pdf).

Additional comments

I will include here only the most important comments that need an urgent revision. But there are much more suggestions, comments and requestings of changes in the annotated pdf, that should also be considered by the authors.

Lines 33-36. Mesosaur partitioning: By what evidence you support these statements? Limestones and shales can characterize either shallow or deeper environments in different settings of the Paraná and Karoo basins. Juveniles are not more common in limestones, in Uruguay for instance, they are in the shale and also in the siltstones and dolomite facies, as well as the adults.
However, juvenile mesosaurs could be more commonly found in presumed shallow environments, because their food is in the shore, as was previously suggested (see for instance Silva et al. 2017; Núñez Demarco et al., 2018). Regarding the more articulated specimens that seem to be preserved in the deepest areas of the basins not all are adults, some are juvenile or subadult individuals. Lastly, you can find also completely disarticulated skeletons in the limestones including a mixture of large adults and juvenile and very young specimens (see Piñeiro et al., 2012a-b).
Line 74. Mesosaur Dietary: Coprolites, and cololites plus regurgitalites are not exclusive of the Iratí Fm., they are well represented in the Mangrullo Fm. (see Piñeiro et al. 2012b and Silva et al., 2017). Indeed, gut contents are only preserved in the Mangrullo Fm. (Piñeiro et al., 2012b). So, please refer the research of other colleagues adequately, taking into account specifically authorship.
Lines 92-94. Mesosaur Dietary and preferencial environments: First of all, there are no controversial issues about the mesosaur diet and environments since gut contents, coprolites, cololites and regurgitalites have been described and provided enough evidence of prey preferences of a tetrapod which is the only present in the deposits of the Iratí and Mangrullo formations where these fossils were found (Piñeiro et al. 2012b; Ramos, 2015; Silva et al. 2017). Moreover, mesosaurs are not marine animals as was demonstrated in the last years, they lived in continental seas that became lakes with variable salinity and hypersaline lagoons (Piñeiro et al. 2012b). Also, we have evidence now that seem to support previous hypotheses that mesosaurs could have had aquatic ancestors (Nuñez Demarco et al. 2022), a paper that you will like to see and refer in this manuscript if it is finally accepted). Therefore, this sentence has no sense.
Lines 98-101. Taxonomy and niche partitioning: This paragraph is confusing. Firstly, there is no evidence that support niche partitioning as I commented above and secondly, if you found few but significant differences in osteological and dental proportions between specimens that you assigned to different taxa, what led you to consider them as just one species in your conclusions? Please, explain.
Lines 304-307. Taxonomy and ontogeny: Again, this is unsupported by the data available that have demonstrated that there is no evidence of three mesosaur types. Your “Brazilosaurus” type is not a poorly preserved mesosaur, and your purported “Stereosternum type” include several adults, more than juveniles, and the Mesosaurus type that you mention is represented by several juveniles, even by one still unborn individual! Therefore, the evidence support that there is only one species (one type) represented by several ontogenetic stages; a very unusual preservation for a species that lived near 300 million years ago. If you have morphological and anatomical characters from which recognize unambiguously more than one mesosaur taxon, please describe them here. Taphonomic characters are not useful to solve taxonomy.
Lines 312-317. Geology, biostratigraphy and sedimentology of the mesosaur-bearing deposits: Completely wrong and unsupported statements you include in this paragraph. All mesosaurs are in correlated lithostratigraphic units, they are not in geological different strata. Mesosaur remains are not restricted to the black shales or the limestones, they are also in silstones and even in some basalmost mudstones. Besides, you do not provide real anatomical characters to recognize different mesosaurs in each facies. The mesosaur species that is found at the limestones is Mesosaurus, which is also represented in the shale, siltstones and mudstones and dolomite! You have to make some field work to the deposits containing mesosaurs and perform serious taphonomic studies before to suggest that there can be niche partitioning in mesosaurs. As I have already mentioned, the only taphonomic feature that deserve to be mentioned are the abundant preservation of mostly complete or partially articulated skeletons in the shale or limestone from the deepest part of the basins, which is probably due to the absence of scavengers during biostratinomic putrefaction, bioturbating the dead bodies in the saline and anoxic bottoms. The opposite is the presence of “bone beds” or lens of completely disarticulated specimens in the shallowest zones (coastal) of the large sea or lagoon where mesosaurs lived, where the conditions are more variable and where there is a more complex community including crustaceans and insects, and plants. These aspects have already been well studied by previous authors and you can explore them to increase your knowledge about the mesosaur-bearing deposits (see Soares, 2003; Piñeiro 2006, 2008; Piñeiro et al. 2012a-c; Silva et al., 2017; Nuñez Demarco et al. 2018; among many others!).
Lines 323-325. Methodology and Discussion not matching: I can see that you have more significant differences in your statistical results than those that were used previously to support the three mesosaur taxa. But you do not consider them as significant to discuss, in the light of your consideration about the existent of a single mesosaur species in support to previos results. Instead, you wasted two pages of the discussion by looking for ecological and developmental evidence that do not explain these differences. So, you concluded that even though the differences exist, they are not significant, but you do not explain why you discharge them. Thus, it seems that you discuss on aspects that you knew with anticipation that will be discharged by your statistical results. Thus there is a redundant circular discussion. The Discussion section should be the most important of your manuscript and it should be based on the results obtained by the applied methodology, which as I understand, found some differences between the studied specimens assigned to Mesosaurus with respect to those assigned to the other two taxa. Therefore, it will be better that you discus the differences that you found between the examined specimens regarding previous contributions that also found perhaps similar variation. On the other hand, you can discuss about taphonomy and environmental and ecological preferences, but leaving clear that you are considering your own results (in agreement or not with previous work) and taking into account the conditions present in the complete distribution of mesosaurs.
Lines 335-343. Morphometrics: What you say in this section may be a possibility regarding one single taxon preserved in several ontogenetic stages, but you need to study more juveniles (very young and more aged ones) and compare the size of manus and pes on them. Doing so, you will acknowledge that your results are wrong. Manus is always smaller than pes, even in the unborn individuals. Such difference in size is maintained during the juvenile and adult stages, but it becomes less accentuated when they become adults.

Lines 344-348. More on niche partitioning: Your evidence for supporting niche portioning is not correct.
Concerning your proposed dietary partition also you do not present any evidence. Pygocephalomorph crustaceans as the most important item in the mesosaur diet were already studied by Pinto and Adami-Rodriguez, 1996; Matos et al. 2013; Ramos, 2015, Adami-Rodriguez et al., 2016; and Silva et al. 2017 among others. See these contributions and the papers cited therein. But one thing is very important to clarify here and it is how you recognize a specimen that is a juvenile and other that is a subadult or a young adult to support your hypothesis that juveniles and adults live in separate environments? Fossil record indicates the entire contrary.
Line 367. Konservat Lagerstätte of the Mangrullo Formation: Do you think that the preservation of an embryo inside an egg and a pregnant female, along to mandibles with the trigeminal nerve preserved as a phosphatized structure, gastric contents and coprolites and the presence of salt glands and their conducts are no soft tissue preservation that should attract your attention for your analysis. Why you use as a reference for preservation of soft tissue in mesosaurs only one paper that is not specific?

Lines 390-400. Final conclusions: Your conclusions could be interesting but the evidence available does not support them. Adult and juvenile mesosaurs are found in association in Uruguay (see Piñeiro et al. 2012a), Brazil (see Piñeiro et al. 2012a) and also in Africa (e.g., Namibia, see Piñeiro et al. 2021). Thus, you need to provide some data that support your hypotheses, for instance, you could find evidence that there is a niche partitioning in a particular locality and thus you have to discuss the factors that led to such a distribution. But people that have collected the materials that are now studying have worked with mesosaurs at the field for more than 20 years and have observed other conditions that are described in their papers. You cannot ignore their work.

In sum I think that you have to revise your methodology and according with the results that you obtain prepare a new discussion having into account this review and of course the information previously published by other colleagues. You cannot present a new taphonomic or palaeoecological model for mesosaurs taking into account just the most beautiful preserved specimens that were sold to the museums. I am delighted to revise a new version of this manuscript that I hope will be improved by the inputs provided by the Reviewers and the Handling Editor.

Sincerely,

Graciela Piñeiro

·

Basic reporting

- Clear, unambiguous, professional English language used throughout. (Yes)
- Intro & background to show context. (Yes)
- Literature well referenced & relevant. (Not quite)
Some additional literature was recommended.
- Structure conforms to PeerJ standards, discipline norm, or improved for clarity. (Yes)
- Figures are relevant, high quality, well labelled & described. (Yes)
- Raw data supplied (see PeerJ policy). (Not quite)
The number of specimens shown in Table 1 does not correspond to the total number of specimens reported in the text.

Experimental design

- Original primary research within Scope of the journal. (Yes)

- Research question well defined, relevant & meaningful. It is stated how the research fills an
identified knowledge gap. (Yes)

- Rigorous investigation performed to a high technical & ethical standard. (Yes)

- Methods described with sufficient detail & information to replicate. (Yes)

Validity of the findings

- All underlying data have been provided, are robust, statistically sound, & controlled. (Not quite)
It was missing to discuss about the records of adults and juvenile specimens found associated (mentioned in the literature).

- Conclusions are well stated, linked to original research question & limited to supporting
results (Not quite)
The conclusions are perfectly coherent with the analyzed data. However, I understand that some other data should have been considered (see previous comment).

Additional comments

It is a very interesting paper and deserves to be published, with minor changes required.
The discussion regarding the taxonomy of mesosaurs is old and still controversial. The novelty brought by the authors (which corresponds to the strong point of the article) is the use of morphometric techniques to perform a comparison between specimens from different places. The analyzed sample is sufficiently robust to give credibility to the results obtained. My only caveat in relation to the conclusions obtained refers to the lack of discussion regarding the existence of adult and juvenile specimens found associated (that is mentioned in the literature consulted by the authors).

*Additional comments were made in the attached .PDF.

---

## Round 0.2 · Major Revisions

Our first reviewer made some further suggestions that need your attention. Please explain why you did not include specimens from the Mangrullo Formation of Uruguay and the Paleontological Museum of the Zoobotanic Foundation in Porto Alegre, Brazil. It would help if you described the acronyms for the institutions from which the studied specimens come. Please clarify whether the materials used to analyze the growth pattern for the mesosaur skull (i.e., PIMUZ III 192, 513; SMF-R 387, 4512, 4513) are suitable for that purpose. Also, clarification is needed concerning the specimen USNM-V-41249 analyzed. About the characterization of the invalidated taxa, you should consider the almost complete specimens with even soft tissues preserved that were arbitrarily assigned to “Brazilosaurus”.

Our second reviewer suggests adding a figure illustrating the statement that Brazilosaurus specimens are misidentified based on taphonomically displaced ribcages. I also think that this figure could make your assertions more rigorous.

·

Basic reporting

This is a manuscript that I have revised at its first stage of submission and for which I have marked many concerns. I realize that the new version of the manuscript although very modified has yet many severe problems. I will describe them in detail below, within the Experimental Design and Validity of the Findings sections.
The literature is now adequate but authors should refer the credits of contributions to the corresponding owners along the body text.
Figures should be improved. At least the files that I received for review are of poor quality and the graphics can be improved as they do not allow distinguishing the specimens used in the study. Moreover, additional figures have to be provided, particularly for to support many of the morphometric and statistical new results obtained (see below).

Experimental design

Although the new version of the manuscript has substantially changed with respect to the first submitted, definitively this is not original primary research. This manuscript continues to be an effort to test and eventually support or discredit previous articles in the subject (i.e., Piñeiro et al., 2021; Nuñez Demarco et al., 2022), reproducing the same methodology. Of course, that is completely valid in science as long as the credits for the original publications are indicated.
The main research questions are not clearly defined in this manuscript because even the title masks the real objectives which are based in the application of morphometric analyses. At least something referring to that methodology should be displayed in the title.
Taking into account that a worldwide study of mesosaurs is claimed to be here performed, and it is much unexplained that materials from the Mangrullo Formation of Uruguay were not included, even when the results that are being revised in this manuscript were based in the study of fossils coming from such formation. Other discharged paleontological collection for this “worldwide” study is the Paleontological Museum of the Zoobotanic Foundation in Porto Alegre, Brazil, which houses many of the specimens used to perform the analyses in Piñeiro et al., 2021 and Nuñez Demarco et al., 2022. I would like to see an explanation for the reasons that led author to discharge these fossils.

Goals and ethics

Among the goals described by the authors, some of them seem to be assigned to their own, but they are results already provided in previous several papers. Other results are not supported by the fossils or their taphonomy, and are also based on questionable geological, and environmental considerations elaborated only from samples of the paleontological collections.
For instance, in the new abstract authors stated: “This data presents evidence of surprising ontogenetic changes in these animals as well as new insights into their taxonomy. Our results support the recent hypothesis that Mesosaurus tenuidens is the only valid species within Mesosauridae and suggest that “Stereosternum tumidum” and “Brazilosaurus sanpauloensis” represent immature stages or incomplete specimens of Mesosaurus by showing that all three species occupy an incomplete portion of the overall size range of mesosaurs.”

These results claimed by the authors as the main insides for their manuscript (e.g., “The “new insights” we mention here are the evidence-based consideration that “Brazilosaurus” and “Stereosternum” might represent a poor state of preservation and a range of immature ontogenetic stages of Mesosaurus, respectively. To our knowledge, this has not been proposed in any previous publication.” all are ideas previously proposed by different authors differently written.

See as an example the following paragraphs extracted from Piñeiro et al. (2021), who wrote at page 232:

“Among the previous workers that supported the presence of just one taxon of mesosaurs in the Paraná and the Karoo basins are Friedrich von Huene (1940, 1941) and Alfred S. Romer (1956). These brilliant anatomists considered Mesosaurus tenuidens as the only mesosaur species and believed that the subtle differences observed were just related to the ontogenetic stage of development of the preserved individuals.”

Also, see the following paragraph in Piñeiro et al. 2021:

“The anatomical conditions that were previously used to support the presence of three monotypic mesosaur taxa (apud Araújo, 1977; Modesto, 1996, 1999, 2010) were here statistically analyzed and our results suggest that they are natural intraspecific variations produced during the individual ontogenetic growth and driven by specific environmental or ecological stress.”

Furthermore, there are more previous references to the proposed “new insights” of the manuscript in review:

"...the morphological differences that would assure the recognition of the three monotypic mesosaur taxa are indeed very weak. Piñeiro (2002, 2004) and more recently Piñeiro et al. (2012a, b; 2016) and Laurin & Piñeiro (2017, 2018) have questioned most of such differences arguing that they are derived from taphonomy and may represent ontogenetic and intraspecific variability..."
(Piñeiro et al., 2021, p.207).

And even more:

"The contributions from the last two authors suggest that most of the characters used to separate mesosaur species may be associated to ontogenetic changes and/or taphonomic artefacts and that Mesosaurus is the only mesosaur taxon present in Uruguay (Morosi 2011)"
(Piñeiro et al 2012, p.2).

Therefore, if the selected text is maintained in the abstract, it must be clarified that these statements are recognitions of results suggested in previous papers.

Methodology and Results

Many concerns that I have already remarked in my first review still remain in the methodological section.
A description of the acronyms for the institutions from which the studied specimens come should be provided.
The performed analyses are modified from those used in a paper recently published by my research group (Nuñez Demarco et al. 2022); the same sections of the skeleton, the same bones, the same measures and similar statistics, and the same characteristics of these bones are a surprising replication of that paper. In this regard and despite the declaration of the authors in the rebuttal letter that it is a coincidence given that the performed studies for this manuscript are related to a project research that they have since some years ago, they would explain that they are using similar methodological approaches in order to revise the results recently obtained by Nuñez Demarco et al. 2022, who found that mesosaurs developed an isometric growth pattern which was detected before only in recumbirostran lepospondyls, among amniotes. This is important for the readers to compare both the results obtained by applying similar methodologies, and also to discuss the hypothesis that mesosaurs are secondary aquatic animals. If it is not possible to provide evidence for this statement it is needed to be cautious and consider all the hypotheses (e.g., Romer, 1947; Piñeiro et al. 2022), in particular those that are supported by relevant evidence , even if authors now have changed all their statistical results and recall that isometry in mesosaurs is produced by the analysis of a smaller sample of 109 specimens in comparison with their 270 (which indeed are 268 because specimens preserved in parts and counterparts were considered twice.
Obviously, as large the sample is, the possibilities to have more accurate results would increase, but while in this manuscript in review any specimen to enlarge the sample was accepted, in our sample only the best-preserved specimens were selected in order to avoid commit mistakes in the delimitation of the bones or body regions to be measured.
Curiously, I revised the list of specimens provided in S Table 1 and realized that for instance, some materials used to analyze the growth pattern for the mesosaur skull (i.e., PIMUZ III 192, 513; SMF-R 387, 4512, 4513) are not suitable for that purpose because they do not allow obtaining an exact measurement of the snout, which is broken or incompletely preserved. The last one even is bad to measure the postorbital region length. I really have concerns about the preservation condition of the other eighty skulls from which they suggest an allometric pattern. I would suggest authors to identify the specimen numbers where the measurements for the skulls were inferred and also recommend to provide two studies one, including all the specimens with skull is available but the measurements, particularly those from the snout are inferred and other for examine the skulls where all the measures taken are exact. Providing better graphics can help readers to individualize the specimens plotted in each study.
There are even other problematic specimens: the specimen USNM-V-412494, (which according to the Museum online database is USNM PAL 412494) is actually a group of 9 different specimens. Thus it should be indicated which one was studied. In the specimen USNM V 25859, they measured the humerus length but the humerus is not complete, the proximal section is missing.

Reduction of the fore and hind limbs

It is stated at lines 283-295 of the revised manuscript that the performed studies indicate that there is a reduction of the mesosaur hind limb and a reduction of the manus during ontogeny, considering the one species hypothesis of Piñeiro et al., 2021. Also, authors declare that this result is very different from that obtained in Nuñez Demarco et al. 2022. But they should explain to respect to what they see that there is a reduction of the limbs in mesosaurs. According to the data provided in the Materials and methods section, it seems that they are considering the mean of the dorsal vertebrae length as a comparative measure. If so, the results obtained are very different to those obtained by Nuñez Demarco et al. (2022) because in this study all the measured bones were compared to the mean of the dorsal vertebral length and Nuñez Demarco et al. (2022) instead used the mean of all the available vertebra, not only the dorsal vertebrae.
Moreover, it is important to take into account the intraspecific differences existing within the populations that even can be related to the variable environmental conditions that could have been present along the very extended Paraná and Karoo basin. As they considered a large sample, the intraspecific differences in growing and even in development of individuals of similar age, could constrain the interpretation of the different results, which otherwise are not significant.
In this respect, some allometric values are very close to zero and the error intervals seem to include zero. How noticeable is this size variation? How significant is this allometry in the anatomy of a mesosaur? because these values are very close to isometry, and in practice, with these error intervals, the observed allometry seems to be almost imperceptible.

It was also informed in the rebuttal letter that “it was statistically tested whether the value of each allometry coefficient was significantly different from zero using t-tests, which proved even more conservative than simple confidence interval comparisons” (I would like to see a reference for this statement). The results presented in Tables 1-3 are referred, where the ontogenetic stage of the analyzed specimens is not provided, but only the different mesosaur “taxa” that appear as still supported. However, this study, as well as other theoretical hypotheses provided, is not consistent with the reality because they are comparing two virtual taxa with a real taxon. Otherwise, if the ontogenetic stage of the measured specimens is rather provided, meaning that a subadult (“Brazilosaurus”) and a juvenile (“Stereosternum”) are compared to an adult Mesosaurus, the results should be equivalent for the three cases or instead the eventually different values could represent intraspecific variation within Mesosaurus. Nevertheless, the tables provided show significant differences that seem inexplicable if just one taxon (even including different ontogenetic stages) is being studied. Even so, it is declared that the single species hypothesis of Piñeiro et al. 2021 is supported. This is the main ambiguity that I detected since the former revision of this manuscript.

Therefore, the section corresponding to the “Limbs” should be modified accordingly. Moreover, I would be happy to see a figure including at least one juvenile, one subadult and one adult mesosaur from the large number that you revised (and I imagine you have taken photographs) to compare the evolution of the fore and hind limb proportions directly from the specimens.

The Iratí-Whitehill “sea” of the authors excludes the very fossiliferous deposits from the Mangrullo Formation from Uruguay without providing any rational explanation for that.

Apparently, it was decided to restring this study to the Iratí Fm. (Paraná basin) in Brazil and the Whitehill Fm. (Karoo basin) in Africa excluding the correlative Uruguayan deposits without an explanation. This is a serious problem from my point of view because reconstructing of the environmental conditions suggested for support the partitioning hypothesis would be better tested with the information about all the basins representing the entire water body. Moreover, the Mangrullo Formation and its fossils have promoted several novel papers in the last years, some of which are in the main focus of this manuscript.
A marine environment is assigned for what is called the Iratí-Whitehill Sea. But the most accepted hypothesis is that the water body occupied by mesosaurs and pygocephalomorph crustaceans is an epeiric sea (Xavier, 2018). I think that authors misinterpret an epeiric sea as equivalent to an ocean. But it is a body of water of variable salinity and environments that develops within a continent, it can be an intermittent connection with the open ocean and so, the epeiric sea may be comparable to a big lake with very variable and complex environmental conditions, where marine communities cannot develop. “Fish” remains, which were abundant at the Taquaral Member of the Iratí Fm and at the basal mudstones of the Mangrullo Fm. practically disappear after the establishment of the restricted conditions in the basins and despite very sporadic disarticulated scales can appear in some localities as the Passo de São Borja in Southern Brazil (Xavier, 2018) they are reworked elements found in a light-grey siltstone core. Pedro Xavier has explained that “there is no conspicuous association of "fish" and mesosaurs and pygocephalomorphs; only one isolated scale mixed with many disarticulated carapaces of small pygocephalomorphs was found in Passo de São Borja locality” (Pedro Xavier, personal communication, 2022).

“Niche and diet partitioning” hypotheses are still claimed

Mostly articulated specimens, almost complete and more or less well-preserved individuals are interpreted as only preserved in limestones. Also, from the histograms shown at figures 6 and 7 it can be seen that very immature individuals (vertebral size less than 5mm, I suppose that is vertebral length, but it is not specified) are better represented in the limestone and so, they are restricted to the shallower environments. However, there are also young adults and subadults in the limestone, which indeed can be represented by a mix of carbonate rocks such as mudstone, grainstone and packstone according to some authors). On the other hand, very immature juveniles seem to be almost absent in the shale, being black bituminous or light, meaning that mostly adult mesosaurs visited the deepest areas of the water body. Even considering the scenario shown by the histograms, the taphonomic frame for mesosaurs is substantially more complex, and it is not possible to suggest a taphonomic model based on analyses of fossils in collections. It is possible that juvenile mesosaurs are overrepresented in some types of limestone because these sedimentary rocks favor the preservation of the fossils. There could have been an equivalent number of adults or subadults in the limestone, but there was a bias in the collecting, the workers at the quarries obviously selected the more attractive specimens but others “badly preserved” (e.g., disarticulated bones or just parts of the skeleton) that could have been closely associated were not considered useful to promotion and were discharged. This can be the reason for the very little information available about them at the museums and institutions around the world. That bias cannot be noted when a scientific collection is revised, because it consists of specimens showing several taphonomic categories for adults, and both subadult and juvenile individuals. A good example can be found in the São Paulo and Rio de Janeiro collections as well as in those housed at the Zoobotanic Foundation of Porto Alegre and at the Facultad de Ciencias of Montevideo, Uruguay.
At the light shale and siltstone both juveniles and adults and also subadults can be found, but the completeness is in general lower than in the limestone, although the preservation of delicate structures is astonishing in natural molds. In the Mangrullo Formation we have found some more complete specimens in the shale, including one practically complete juvenile along with a well-preserved adult individual. Also, at the shale and siltstone, fossils of large pygocephalomorphs and plants and even insects suggest a community preserved in rocks that were deposited in an environment close to the coastal area.
I think that all these things should be discussed.

Concerning the juvenile mesosaur diet as being different to that of the adults, I would like to see the proper evidence that the authors have for supporting such a hypothesis. Perhaps some of the studied juvenile specimens preserved gastric contents or coprolites with bitten pygocephalomorphs?
Once again, I should advise that “fish” remains are not found in levels with mesosaurs and pygocephalomorphs, but if you have any proof of such association, please, present it as evidence because that will be in fact a novel discovery.

Discussion

The invalidated mesosaur taxa are again the focus of the discussion, where each “taxon” is characterized as it was defined for the type specimens! Thus, it is suggested that “the few “Brazilosaurus” specimens studied here are concentrated in the adult size range of “Stereosternum”, but no juveniles or subadults are preserved. Second, there is a clear divide between the youngest “Stereosternum” specimens on the one hand and subadults and adults of that species on the other hand. Third, and more importantly, only two specimens of Mesosaurus are the size of juvenile “Stereosternum”, whereas all other specimens range from the size of “Stereosternum” subadults to larger sizes than the largest “Stereosternum” specimens.”
The specimens that they are talking about are all Mesosaurus, they were arbitrarily assigned to Brazilosaurus and Stereosternum. Therefore, how could one characterize a subadult Stereosternum if it is a subadult Mesosaurus? Then it is a confusing and ambiguous way to discuss what was already discussed and resolved.

The final of the discussion is also very debatable: “Our results therefore corroborates the single-species hypothesis of Piñeiro et al. (2021) and their discarding of the former characters as ontogenetically variable. In fact, the three previously identified “species” more probably represented three artificial types: the “Brazilosaurus”-type is a poorly preserved mesosaur with displaced ribs, the “Stereosternum”-type is a size class encompassing juveniles to young adults, and the “Mesosaurus”-type represents adults with more extreme sizes and morphologies. Consequently, hereafter we will only consider our results in the light of the single-species hypothesis.”

This is an incorrect and unreal characterization of the invalidated taxa. There are beautiful almost complete specimens preserving even soft tissues that were arbitrarily assigned to “Brazilosaurus”, “Stereosternum” can’t be any ontogenetic stage of Mesosaurus because all mesosaur specimens represent this last taxon.

Thus, my conclusion is that many of the paragraphs at the discussion should be removed to avoid wrong conjectures.

Validity of the findings

The main value of this article is the revision of the results presented in Piñeiro et al. 2021 about the identification of just a single species of mesosaurs which is Mesosaurus tenuidens and on the other hand to test the results provided by Nuñez Demarco et al. (2022) about the isometry as the main growth pattern in mesosaurs, a pattern only present in recumbirostran lepospondyls. Other taphonomic and environmental hypotheses presented in this manuscript cannot be totally validated because of the lack of well-detailed studies on the mesosaur-bearing deposits made directly from the field. Due to the poor knowledge of the basic stratigraphy, sedimentology and paleoecology of these deposits the conclusions should be taken with caution.

·

Basic reporting

The text is well written. Minor suggestions were made to the text file, which is sent to the authors.
My only major suggestion is to add a figure illustrating the statement tht Brazilosaurus specimens are misidentified based on taphonomically displaced ribcages. This is a very interesting observation, not only because this gives base to the authors when they merge all mesosaurid taxa together, but also because Brazilosaurus specimens are overall poorly depicted in the literature and, therefore, very problematic. Personally, I also saw Stereostenum and Mesosaurus as very similar taxa, especially theirs postcrania. On the other hand, for me Brazilosaurus had more distinctive traits (especially their putatively long necks).
If this is a misinterpretation, as the authors observed, providing photographs that illustrate such claims will not only enrich the text, reinforce the merging of the taxonomic groups, and thus encourage further citations of this work.

Experimental design

No comment.

Validity of the findings

No comments.

Additional comments

This is an importnt study, and was an interesting reading. I am pleased to see mesosaurs gaining attention in academic research in the latter years, since this is a very enigmatic, well sampled, but paradoxically neglected group of reptiles. The implications discussed in the text explore a thought-provoking aspect of mesosaur paleoecology, i.e. the correlation between facies and morphotypes/taxa; and the possibility of niche-partitioning among said morphotypes. The statistic treatment given by the authors to the data poses a compelling question to the validity of the three mesosaur genera. I was originally convinced of the existence of these three taxa, but this work makes me rethink this. Though the settling of this matter, for me, still requires a detailed anatomical assessment of mesosaur ontogeny and/or taxonomy (which is outside the objectives of this paper), the data and methods explored by the authors bring important facts (and questions) to such a neglected group of vertebrates.

---

## Round 0.3 · accepted · Accept

I have noted the disagreement between the authors and one of the reviewers. However it Is my opinion that the authors have revised appropriately and therefore I believe this article can now be Accepted, allowing the wider community to form its own opinions on the published article.